# How to fix a broken confidence estimator: Evaluating post-hoc methods for selective classification with deep neural networks

## Abstract

This paper addresses the problem of selective classification for deep neural networks, where a model is allowed to abstain from low-confidence predictions to avoid potential errors. We focus on so-called post-hoc methods, which replace the confidence estimator of a given classifier without retraining or modifying it, thus being practically appealing. Considering neural networks with softmax outputs, our goal is to identify the best confidence estimator that can be computed directly from the unnormalized logits. This problem is motivated by the intriguing observation in recent work that many classifiers appear to have a "broken" confidence estimator, in the sense that their selective classification performance is much worse than what could be expected by their corresponding accuracies. We perform an extensive experimental study of many existing and proposed confidence estimators applied to 84 pretrained ImageNet classifiers available from popular repositories. Our results show that a simple $p$-norm normalization of the logits, followed by taking the maximum logit as the confidence estimator, can lead to considerable gains in selective classification performance, completely fixing the pathological behavior observed in many classifiers. As a consequence, the selective classification performance of any classifier becomes almost entirely determined by its corresponding accuracy. Moreover, these results are shown to be consistent under distribution shift. We also investigate why certain classifiers innately have a good confidence estimator that apparently cannot be improved by post-hoc methods.

## 1 Introduction

Consider a machine learning classifier that does not reach the desired performance for the intended application, even after significant development time. This may occur for a variety of reasons: the problem is too hard for the current technology; more development resources (data, compute or time) are needed than what is economically feasible for the specific situation; or perhaps the target distribution is different from the training one. In this case, then, should the model *not* be deployed?

Selective classification Geifman & El-Yaniv (2017); El-Yaniv & Wiener (2010) offers a potential solution, which can be seen as a last resort. The idea is to reject predictions for which the model is least confident, hoping to increase the performance on the accepted predictions. The rejected inputs may be processed in the same way as if the model were not deployed, for instance, by a human specialist or by the previously existing system. This imposes a burden on the deployed model, which is run for all inputs but is useful only for a subset of them. Still, it offers a tradeoff between performance and *coverage* (the proportion of accepted predictions) which may be a better solution than any of the extremes. In particular, it could shorten the path to adoption of deep learning especially in critical applications, such as medical diagnosis and autonomous driving, where the consequences of erroneous decisions can be severe (Zou et al., 2023; Neumann et al., 2018).

A key element in selective classification is the confidence estimator that is thresholded to decide whether a prediction is accepted. In the case of neural networks with softmax outputs, the natural baseline to be used as a confidence estimator is the maximum softmax probability (MSP) produced by the model, also known as the softmax response (Geifman & El-Yaniv, 2017; Hendrycks & Gimpel, 2016). Several approaches have been proposed attempting to improve upon this baseline, which

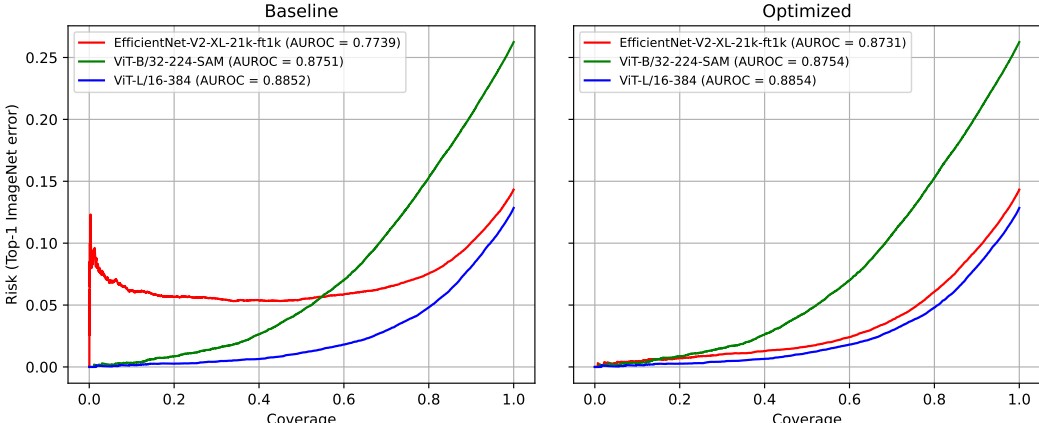

Figure 1: A comparison of RC curves made by three models selected in (Galil et al., 2023), including examples of highest (ViT-L/16-384) and lowest (EfficientNet-V2-XL) AUROC. An RC curve shows the tradeoff between risk (in this case, error rate) and coverage. The initial risk for any classifier is found at the 100% coverage point, where all predictions are accepted. Normally, the risk can be reduced by reducing coverage (which is done by increasing the selection threshold); for instance, a 2% error rate can be obtained at 36.2% coverage for the ViT-B/32-224-SAM model and at 61.9% coverage for the ViT-L/16-38 model. However, for the EfficientNet-V2-XL model, this error rate is not achievable at any coverage, since its RC curve is lower bounded by 5% risk. Moreover, this RC curve is actually non-monotonic, with an increasing risk as coverage is reduced, for low coverage. Fortunately, this apparent pathology in EfficientNet-V2-XL completely disappears after a simple post-hoc optimization of its confidence estimator, resulting in significantly improved selective classification performance. In particular, a 2% error rate can then be achieved at 55.3% coverage.

generally fall into two categories: approaches that require retraining the classifier, by modifying some aspect of the architecture or the training procedure, possibly adding an auxiliary head as the confidence estimator (Geifman & El-Yaniv, 2019; Liu et al., 2019a; Huang et al., 2020); and post-hoc approaches that do not require retraining, thus only modifying or replacing the confidence estimator based on outputs or intermediate features produced by the model (Corbière et al., 2022; Granese et al., 2021; Shen et al., 2022; Galil et al., 2023).[1]

Post-hoc approaches not only are practically appealing, especially for large deep learning models, but also should serve as a baseline against which to compare any training-based approaches. However, a thorough evaluation of potential post-hoc estimators has not yet appeared in the literature. The work furthest in that direction is the paper by Galil et al. (2023), who empirically evaluated ImageNet classifiers and found that temperature scaling (TS) (Guo et al., 2017), a well-known post-hoc calibration method, could sometimes improve selective classification performance.

In this paper, we focus on the simplest possible class of post-hoc methods, which are those for which the confidence estimator can be computed directly from the network unnormalized *logits* (pre-softmax output). Our main goal is to identify the methods that produce the largest gains in selective classification performance, measured by the area under the risk-coverage curve (AURC); however, as in general these methods can have hyperparameters that need to be tuned on hold-out data, we are also concerned with data efficiency. Our study is motivated by an intriguing problem reported in (Galil et al., 2023) and illustrated in Fig. 1: some state-of-the-art ImageNet classifiers, despite attaining excellent predictive performance, nevertheless exhibit appallingly poor performance at detecting their own mistakes. Can such pathologies be fixed by simple post-hoc methods?

To answer this question, we consider every such method to our knowledge, as well as several variations and novel methods that we propose, and perform an extensive experimental study using 84 pretrained ImageNet classifiers available from popular repositories. Our results show that, among other close contenders, a simple $p$-norm normalization of the logits, followed by taking the maximum logit as the confidence estimator, can lead to considerable gains in selective classification performance,

---

[1]For more complete account of related work, please see Appendix A.

completely fixing the pathological behavior observed in many classifiers, as illustrated in Fig. 1. As a consequence, the selective classification performance of any classifier becomes almost entirely determined by its corresponding accuracy.

In summary, the contributions of this work are:

- We propose a simple but powerful framework for designing confidence estimators, which involves tunable logit transformations optimized directly for a selective classification metric;
- We perform an extensive experimental study of many existing and proposed confidence estimators applied to 84 pretrained ImageNet classifiers available from popular repositories. In particular, we show that simple post-hoc estimators, such as the max-logit normalized by the logit $p$-norm, can provide up to 62% reduction in normalized AURC using no more than one sample per class of labeled hold-out data;
- We study how these post-hoc methods perform under distribution shift and find that the results remain consistent: a method that provides gains in the in-distribution scenario also provides considerable gains under distribution shift;
- We investigate why certain classifiers innately have a good confidence estimator that apparently cannot be improved by post-hoc methods.

## 2 PROBLEM FORMULATION AND BACKGROUND

### 2.1 SELECTIVE CLASSIFICATION

Let $P$ be an unknown distribution over $\mathcal{X} \times \mathcal{Y}$, where $\mathcal{X}$ is the input space and $\mathcal{Y} = \{1, \ldots, C\}$ is the label space, and $C$ is the number of classes. The *risk* of a *classifier* $h : \mathcal{X} \to \mathcal{Y}$ is $R(h) = E_P[\ell(h(x), y)]$, where $\ell : \mathcal{Y} \times \mathcal{Y} \to \mathbb{R}^+$ is a loss function, for instance, the 0/1 loss $\ell(\hat{y}, y) = \mathbb{1}[\hat{y} \neq y]$, where $\mathbb{1}[\cdot]$ denotes the indicator function. A *selective classifier* (Geifman & El-Yaniv, 2017) is a pair $(h, g)$, where $h$ is a classifier and $g : \mathcal{X} \to \mathbb{R}$ is a *confidence estimator* (also known as *confidence score function* or *confidence-rate function*), which quantifies the model's confidence on its prediction for a given input. For some fixed threshold $t$, given an input $x$, the selective model makes a prediction $h(x)$ if $g(x) \geq t$, otherwise the prediction is rejected. A selective model's *coverage* $\phi(h, g) = P[g(x) \geq t]$ is the probability mass of the selected samples in $\mathcal{X}$, while its *selective risk* $R(h, g) = E_P[\ell(h(x), y) \mid g(x) \geq t]$ is its risk restricted to the selected samples. In particular, a model's risk equals its selective risk at *full coverage* (i.e., for $t$ such that $\phi(h, g) = 1$). These quantities can be evaluated empirically given a given a test dataset $\{(x_i, y_i)\}_{i=1}^N$ drawn i.i.d. from $P$, yielding the *empirical coverage* $\hat{\phi}(h, g) = (1/N) \sum_{i=1}^N \mathbb{1}[g(x_i) \geq t]$ and the *empirical selective risk*

$$\hat{R}(h, g) = \frac{\sum_{i=1}^N \ell(h(x_i), y_i) \mathbb{1}[g(x_i) \geq t]}{\sum_{i=1}^N \mathbb{1}[g(x_i) \geq t]}. \tag{1}$$

Note that, by varying $t$, it is generally possible to trade off coverage for selective risk, i.e., a lower selective risk can usually (but not necessarily always) be achieved if more samples are rejected. This tradeoff is captured by the *risk-coverage (RC) curve* (Geifman & El-Yaniv, 2017), a plot of $\hat{R}(h, g)$ as a function of $\hat{\phi}(h, g)$. While the RC curve provides a full picture of the performance of a selective classifier, it is convenient to have a scalar metric that summarizes this curve. A commonly used metric is the *area under the RC curve* (AURC) (Ding et al., 2020; Geifman et al., 2019), denoted by $\mathrm{AURC}(h, g)$. However, when comparing selective models, if two RC curves cross, then each model may have a better selective performance than the other depending on the operating point chosen, which cannot be captured by the AURC. Another interesting metric, which forces the choice of an operating point, is the *selective accuracy constraint* (SAC) (Galil et al., 2023), defined as the maximum coverage allowed for a model to achieve a specified accuracy.

Closely related to selective classification is misclassification detection (Hendrycks & Gimpel, 2016), which refers to the problem of discriminating between correct and incorrect predictions made by a classifier. Both tasks rely on ranking predictions according to their confidence estimates, where correct predictions should be ideally separated from incorrect ones. A usual metric for misclassification detection is the area under the ROC curve (AUROC) (Fawcett, 2006) which, in contrast to the AURC, is blind to the classifier performance, focusing exclusively on the quality of the confidence estimates. Thus, it is also used to evaluate confidence estimators for selective classification (Galil et al., 2023).

## 2.2 CONFIDENCE ESTIMATION

From now on we restrict attention to classifiers that can be decomposed as $h(x) = \arg\max_{k \in \mathcal{Y}} z_k$, where $\mathbf{z} = f(x)$ and $f : \mathcal{X} \to \mathbb{R}^C$ is a neural network. The network output $\mathbf{z}$ is referred to as the (vector of) *logits* or *logit vector*, due to the fact that it is typically applied to a softmax function to obtain an estimate of the posterior distribution $P[y|x]$. The softmax function is defined as

$$\sigma : \mathbb{R}^C \to [0,1]^C, \qquad \sigma_k(\mathbf{z}) = \frac{e^{z_k}}{\sum_{j=1}^C e^{z_j}}, \quad k \in \{1, \ldots, C\} \tag{2}$$

where $\sigma_k(\mathbf{z})$ denotes the $k$th element of the vector $\sigma(\mathbf{z})$.

The most popular confidence estimator is arguably the *maximum softmax probability* (MSP) (Ding et al., 2020), also known as *maximum class probability* (Corbière et al., 2022) or *softmax response* (Geifman & El-Yaniv, 2017)

$$g(x) = \text{MSP}(\mathbf{z}) \triangleq \max_{k \in \mathcal{Y}} \sigma_k(\mathbf{z}) = \sigma_{\hat{y}}(\mathbf{z}) \tag{3}$$

where $\hat{y} = \arg\max_{k \in \mathcal{Y}} z_k$. However, other functions of the logits can be considered. Some examples are the *softmax margin* (Belghazi & Lopez-Paz, 2021; Lubrano et al., 2023), the *max logit* (Hendrycks et al., 2022), the *logits margin* (Streeter, 2018; Lebovitz et al., 2023), the *negative entropy*[2] (Belghazi & Lopez-Paz, 2021), and the *negative Gini index* (Granese et al., 2021; Gomes et al., 2022), defined, respectively, as

$$\text{SoftmaxMargin}(\mathbf{z}) \triangleq \sigma_{\hat{y}}(\mathbf{z}) - \max_{k \in \mathcal{Y}: k \neq \hat{y}} \sigma_k(\mathbf{z}) \tag{4}$$

$$\text{MaxLogit}(\mathbf{z}) \triangleq z_{\hat{y}} \tag{5}$$

$$\text{LogitsMargin}(\mathbf{z}) \triangleq z_{\hat{y}} - \max_{k \in \mathcal{Y}: k \neq \hat{y}} z_k \tag{6}$$

$$\text{NegativeEntropy}(\mathbf{z}) \triangleq \sum_{k \in \mathcal{Y}} \sigma_k(\mathbf{z}) \log \sigma_k(\mathbf{z}) \tag{7}$$

$$\text{NegativeGini}(\mathbf{z}) \triangleq -1 + \sum_{k \in \mathcal{Y}} \sigma_k(\mathbf{z})^2. \tag{8}$$

Note that, in the scenario we consider, DOCTOR's $D_\alpha$ and $D_\beta$ discriminators (Granese et al., 2021) are equivalent to the negative Gini index and MSP confidence estimators, respectively, as discussed in more detail in Appendix B.

## 3 METHODS

### 3.1 TUNABLE LOGIT TRANSFORMATIONS

In this section, we introduce a simple but powerful framework for designing post-hoc confidence estimators for selective classification. The idea is to take any parameter-free logit-based confidence estimator, such as those described in section 2.2, and augment it with a logit transformation parameterized by one or a few hyperparameters, which are then tuned (e.g., via grid search) using a labeled hold-out dataset not used during training of the classifier (i.e., validation data). Moreover, this hyperparameter tuning is done using as objective function not a proxy loss but rather the exact same metric that one is interested in optimizing, for instance, AURC or AUROC. This approach forces us to be conservative about the hyperparameter search space, which is important for data efficiency.

### 3.1.1 TEMPERATURE SCALING

Originally proposed in the context of post-hoc calibration, temperature scaling (TS) (Guo et al., 2017) consists in transforming the logits as $\mathbf{z}' = \mathbf{z}/T$, before applying the softmax function. The parameter $T > 0$, which is called the temperature, is then optimized over hold-out data.

---

[2]Note that any uncertainty estimator can be used as a confidence estimator by taking its negative.

The conventional way of applying TS, as proposed in (Guo et al., 2017) for calibration and referred to here as TS-NLL, consists in optimizing $T$ with respect to the negative log-likelihood (NLL) (Murphy, 2022). Here we instead optimize $T$ using AURC and the resulting method is referred to as TS-AURC.

Note that TS does not affect the ranking of predictions for MaxLogit and LogitsMargin, so it is not applied in these cases.

### 3.1.2 Logit Normalization

Inspired by Wei et al. (2022), who show that logits norms are directly related to overconfidence and propose logit normalization during training, we propose logit normalization as a post-hoc method. Additionally, we extend the normalization from the 2-norm to a general $p$-norm, where $p$ is a tunable hyperparameter. (For more details on the rationale behind logit normalization, please see Appendix C.)

Thus, logit $p$-normalization is defined as the operation

$$\mathbf{z}' = \frac{\mathbf{z}}{\tau \|\mathbf{z}\|_p} \tag{9}$$

where $\|\mathbf{z}\|_p \triangleq (|z_1|^p + \cdots + |z_C|^p)^{1/p}$, $p \in \mathbb{R}$, is the $p$-norm of $\mathbf{z}$ and $\tau > 0$ is a temperature scaling parameter. Note that this transformation is a form of adaptive TS (Balanya et al., 2023), with an input-dependent temperature $\tau \|\mathbf{z}\|_p$.

Logit $p$-normalization introduces two hyperparameters, $p$ and $\tau$, which should be jointly optimized; in this case, we first optimize $\tau$ for each value of $p$ considered and then pick the best value of $p$. Such a transformation, together with the optimization of $p$ and $\tau$, is referred to here as pNorm. The optimizing metric is always AURC and therefore it is omitted from the nomenclature of the method.

Note that, when the underlying confidence estimator is MaxLogit or LogitsMargin, the parameter $\tau$ is irrelevant and is ignored.

## 3.2 Evaluation Metrics

### 3.2.1 Normalized AURC

A common criticism of the AURC metric is that it does not allow for meaningful comparisons across problems (Geifman et al., 2019). An AURC of some arbitrary value, for instance, 0.05, may correspond to an ideal confidence estimator for one classifier (of much higher risk) and to a completely random confidence estimator for another classifier (of risk equal to 0.05). The excess AURC (E-AURC) was proposed by Geifman et al. (2019) to alleviate this problem: for a given classifier $h$ and confidence estimator $g$, it is defined as E-AURC$(h, g) = $ AURC$(h, g) - $ AURC$(h, g^*)$, where $g^*$ corresponds to a hypothetically optimal confidence estimator that perfectly orders samples in decreasing order of their losses. Thus, an ideal confidence estimator always has zero E-AURC.

Unfortunately, E-AURC is still highly sensitive to the classifier's risk, as shown by Galil et al. (2023), who suggested the use of AUROC instead. However, using AUROC for comparing confidence estimators has an intrinsic disadvantage: if we are using AUROC to evaluate the performance of a tunable confidence estimator, it makes sense to optimize it using this same metric. However, as AUROC and AURC are not necessarily monotonically aligned Ding et al. (2020), the resulting confidence estimator will be optimized for a different problem than the one in which we were originally interested (which is selective classification). Ideally, we would like to evaluate confidence estimators using a metric that is a monotonic function of AURC.

We propose a simple modification to E-AURC that eliminates the shortcomings pointed out in (Galil et al., 2023): normalizing by the E-AURC of a random confidence estimator, whose AURC is equal to the classifier's risk. More precisely, we define the normalized AURC (NAURC) as

$$\text{NAURC}(h, g) = \frac{\text{AURC}(h, g) - \text{AURC}(h, g^*)}{R(h) - \text{AURC}(h, g^*)}. \tag{10}$$

Note that this corresponds to a min-max scaling that maps the AURC of the ideal classifier to 0 and the AURC of the random classifier to 1. The resulting NAURC is suitable for comparison across different classifiers and is monotonically related to AURC.

### 3.2.2 MSP FALLBACK

A useful property of MSP-TS (but not MSP-TS-NLL) is that, in the infinite-sample setting, it can never have a worse performance than the MSP baseline, as long as $T = 1$ is included in the search space. It is natural to extend this property to every confidence estimator, for a simple reason: it is very easy to check whether the estimator provides an improvement to the MSP baseline and, if not, then use the MSP instead. Formally, this corresponds to adding a binary hyperparameter indicating an MSP fallback.

Equivalently, when measuring performance across different models, we simply report a (non-negligible) positive gain in NAURC whenever it occurs. More precisely, we define the *average positive gain* (APG) in NAURC as

$$\text{APG}(g) = \frac{1}{|\mathcal{H}|} \sum_{h \in \mathcal{H}} \left[\text{NAURC}(h, \text{MSP}) - \text{NAURC}(h, g)\right]_\epsilon^+, \qquad [x]_\epsilon^+ = \begin{cases} x, & \text{if } x > \epsilon \\ 0, & \text{otherwise} \end{cases}$$

where $\mathcal{H}$ is a set of classifiers and $\epsilon > 0$ is chosen so that only non-negligible gains are reported.

## 4 EXPERIMENTS

All the experiments in this section were performed using PyTorch (Paszke et al., 2019) and all of its provided classifiers pre-trained on ImageNet (Deng et al., 2009). Additionally, some models of the Wightman (2019) repository were utilized, particularly the ones highlighted by Galil et al. (2023). The list of the models, together with all the results per model are presented in Appendix L. In total, 84 ImageNet models were used for experiments. The validation set of ImageNet was randomly split into 5000 hold-out images for post-hoc optimization and 45000 for tests and comparisons. Investigations on the stability of this split are presented in Section F.

To give evidence that our results are not specific to ImageNet, we also run experiments on CIFAR-100 (Krizhevsky, 2009) and Oxford-IIIT Pet (Parkhi et al., 2012) datasets, which are presented in Appendix E.

### 4.1 COMPARISON OF METHODS

We start by evaluating the NAURC of each possible combination of a confidence estimator listed in Section 2.2 with a logit transformation described in Section 3.1, for specific models. Table 1 shows the results for EfficientNet-V2-XL (trained on ImageNet-21K and fine tuned on ImageNet-1K) and VGG16, respectively, the former chosen for having the worst confidence estimator (in terms of AUROC) reported in Galil et al. (2023) and the latter chosen as a representative example of a lower accuracy model with a good confidence estimator.

Table 1: NAURC for post-hoc methods applied to ImageNet classifiers

| Classifier | Confidence Estimator | Logit Transformation | | | |
|---|---|---|---|---|---|
| | | Raw | TS-NLL | TS-AURC | pNorm |
| EfficientNet-V2-XL | MSP | 0.4401 | 0.3504 | 0.2056 | 0.1714 |
| | SoftmaxMargin | 0.3816 | 0.3143 | 0.2033 | 0.1705 |
| | MaxLogit | 0.7695 | – | – | **0.1684** |
| | LogitsMargin | 0.1935 | – | – | 0.1731 |
| | NegativeEntropy | 0.5964 | 0.4286 | 0.2012 | 0.1712 |
| | NegativeGini | 0.4485 | 0.3514 | 0.2067 | 0.1717 |
| VGG16 | MSP | 0.1838 | 0.1850 | 0.1836 | 0.1836 |
| | SoftmaxMargin | 0.1898 | 0.1889 | 0.1887 | 0.1887 |
| | MaxLogit | 0.3375 | – | – | 0.2012 |
| | LogitsMargin | 0.2047 | – | – | 0.2047 |
| | NegativeEntropy | 0.1968 | 0.2055 | 0.1837 | 0.1837 |
| | NegativeGini | 0.1856 | 0.1888 | 0.1837 | 0.1837 |

As can be seen, on EfficientNet-V2-XL, the baseline MSP is easily outperformed by most methods. Surprisingly, the best method is not to use a softmax function but, instead, take the maximum of a $p$-normalized logit vector, leading to a reduction in NAURC of 0.27 points or about 62%.

However, on VGG16, the situation is quite different, as methods that use the unnormalized logits and improve the performance on EfficientNet-V2-XL, such as LogitsMargin and MaxLogit-pNorm, actually degrade it on VGG16. Moreover, the highest improvement obtained, e.g., with MSP-TS-AURC, is so small that it can be considered negligible. (In fact, gains below 0.003 NAURC are visually imperceptible in an AURC curve.) Thus, it is reasonable to assert that none of the post-hoc methods considered is able to outperform the baseline in this case.

In Table 2, we evaluate the average performance of post-hoc methods across all models considered, using the APG-NAURC metric described in Section 3.2.2, where we assume $\epsilon = 0.01$. Figure 2 shows the gains for selected methods for each model, ordered by MaxLogit-pNorm gains. It can be seen that the highest gains are provided by MaxLogit-pNorm, MSP-pNorm, NegativeGini-pNorm and their performance is essentially indistinguishable whenever they provide a non-negligible gain over the baseline. Moreover, the set of models for which significant gains can be obtained appears to be consistent across all methods.

Table 2: APG-NAURC of post-hoc methods across 84 ImageNet classifiers

| Conf. Estimator | Logit Transformation | | | |
|---|---|---|---|---|
| | Raw | TS-NLL | TS-AURC | pNorm |
| MSP | 0.0 | 0.03643 | 0.05776 | 0.06781 |
| SoftmaxMargin | 0.01966 | 0.04093 | 0.05597 | 0.06597 |
| MaxLogit | 0.0 | – | – | **0.06837** |
| LogitsMargin | 0.05501 | – | – | 0.06174 |
| NegativeEntropy | 0.0 | 0.01559 | 0.05904 | 0.06745 |
| NegativeGini | 0.0 | 0.03615 | 0.05816 | 0.06790 |

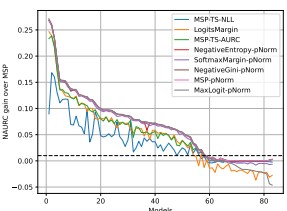

Figure 2: NAURC gains for post-hoc methods across 84 ImageNet classifiers. The dashed line denotes $\epsilon = 0.01$.

Although several post-hoc methods provide considerable gains, they all share a practical limitation which is the requirement of hold-out data for hyperparameter tuning. In Appendix F, we study the data efficiency of some of the best performing methods. MaxLogit-pNorm, having a single hyperparameter, emerges as a clear winner, requiring fewer than 500 samples to achieve near-optimal performance on ImageNet ($< 0.5$ images per class on average) and fewer than 100 samples on CIFAR-100 ($< 1$ image per class on average). These requirements are clearly easily satisfied in practice for typical validation set sizes.

Details on the optimization of $T$ and $p$, additional results showing AUROC values and RC curves, and results on the insensitivity of our conclusions to the choice of $\epsilon$ are provided in Appendix D. In addition, the benefits of a tunable versus fixed $p$, a comparison with other tunable methods, and an analysis of statistical significance are discussed, respectively, in Appendices G, H, I.

### 4.2 POST-HOC OPTIMIZATION FIXES BROKEN CONFIDENCE ESTIMATORS

From Figure 2, we can distinguish two groups of models: those for which the MSP baseline is already the best confidence estimator and those for which post-hoc methods provide considerable gains (particularly, MaxLogit-pNorm). In fact, most models belong to the second group, comprising 58 out of 84 models considered.

Figure 3 illustrates two noteworthy phenomena. First, as previously observed by Galil et al. (2023), certain models exhibit superior accuracy than others but poorer uncertainty estimation, leading to a trade-off when selecting a classifier for selective classification. Second, post-hoc optimization can fix any "broken" confidence estimators. This can be seen in two ways: in Figure 3a, after optimization, all models exhibit roughly the same level of confidence estimation performance (as measured by NAURC), although we can still see some dependency on accuracy (better predictive models are slightly better at predicting their own failures). In Figure 3b, it is clear that, after optimization,

the selective classification performance of any classifier (as measured by AURC) becomes almost entirely determined by its corresponding accuracy. Indeed, the Spearman correlation between AURC and accuracy becomes extremely close to 1. The same conclusions hold for the SAC metric, as shown in Appendix D.5. This implies that any "broken" confidence estimators have been fixed and, consequently, total accuracy becomes the primary determinant of selective performance even at lower coverage levels.

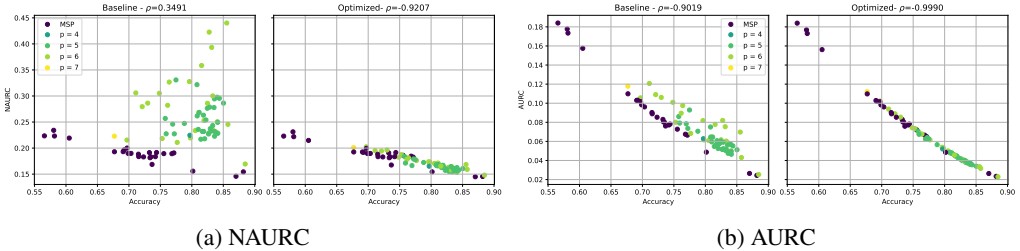

(a) NAURC                                        (b) AURC

Figure 3: NAURC and AURC of 84 ImageNet classifiers with respect to their accuracy, before and after post-hoc optimization. The baseline plots use MSP, while the optimized plots use MaxLogit-pNorm. The legend shows the optimal value of $p$ for each model, where MSP indicates MSP fallback (no significant positive gain). $\rho$ is the Spearman correlation between a metric and the accuracy.

An intriguing question is what properties of a classifier make it bad at confidence estimation. Experiments investigating this question were conducted and presented in Appendix J. In summary, our surprising conclusion is that models that produce highly confident MSPs tend to have better confidence estimators (in terms of NAURC), while models whose MSP distribution is more balanced tend to be easily improvable by post-hoc optimization—which, in turn, makes the resulting confidence estimator concentrated on highly confident values.

## 4.3 PERFORMANCE UNDER DISTRIBUTION SHIFT

We now turn to the question of how post-hoc methods for selective classification perform under distribution shift. Previous works have shown that calibration can be harmed under distribution shift, especially when certain post-hoc methods—such as TS—are applied (Ovadia et al., 2019). To find out whether a similar issue occurs for selective classification, we evaluate selected post-hoc methods on ImageNet-C (Hendrycks & Dietterich, 2018), which consists in 15 different corruptions of the ImageNet's validation set, and on ImageNetV2 (Recht et al., 2019), which is an independent sampling of the ImageNet test set replicating the original dataset creation process. We follow the standard approach for evaluating robustness with these datasets, which is to use them only for inference; thus, the post-hoc methods are optimized using only the 5000 hold-out images from the uncorrupted ImageNet validation dataset. To avoid data leakage, the same split is applied to the ImageNet-C dataset, so that inference is performed only on the 45000 images originally selected as the test set.

First, we evaluate the performance of MaxLogit-pNorm on ImageNet and ImageNetV2 for all classifiers considered. Figure 4a shows that the NAURC gains (over the MSP baseline) obtained for ImageNet translate to similar gains for ImageNetV2, showing that this post-hoc method is quite robust to distribution shift. Then, considering all models after post-hoc optimization with MaxLogit-pNorm, we investigate whether selective classification performance itself (as measured by NAURC) is robust to distribution shift. As can be seen in Figure 4b, the results are consistent, following an affine function; however, a significant degradation in NAURC can be observed for all models under distribution shift. While at first sight this would suggest a lack of robustness, a closer look reveals that it can actually be explained by the natural accuracy drop of the underlying classifier under distribution shift. Indeed, we have already noticed in Figure 3a a negative correlation between the NAURC and the accuracy; in Figure 4c these results are expanded by including the evaluation on ImageNetV2, where we can see that the strong correlation between NAURC and accuracy continues to hold. Similar results are obtained when evaluating classifiers on ImageNet-C, as presented in Appendix K.

Finally, to give a more tangible illustration of the impact of selective classification, Table 3 shows the SAC metric for a ResNet50 under distribution shift, with the target accuracy as the original accuracy obtained with the in-distribution data. As can be seen, the original accuracy can be restored at the

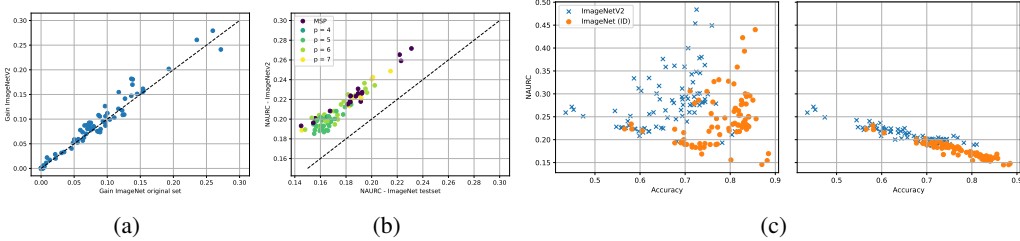

(a)                         (b)                              (c)

Figure 4: (a) NAURC gains (over MSP) on ImageNetV2 versus NAURC gains on the ImageNet test set. (b) NAURC on ImageNetV2 versus NAURC on the ImageNet test set. (c) NAURC versus accuracy. All models are optimized using MaxLogit-pNorm (with MSP fallback).

Table 3: Selective classification performance for a ResNet-50 on ImageNet under distribution shift. For ImageNet-C, each entry is the average across all corruption types for a given level of corruption. The target accuracy is the one achieved for corruption level 0 (i.e., 80.86%).

|  |  | Corruption level | | | | | | |
|---|---|---|---|---|---|---|---|---|
|  | Method | 0 | 1 | 2 | 3 | 4 | 5 | V2 |
| Accuracy [%] | - | 80.86 | 68.56 | 60.03 | 51.85 | 39.44 | 27.09 | 69.97 |
| Coverage (SAC) [%] | MSP | 100 | 71.97 | 52.17 | 37.33 | 19.16 | 8.40 | 76.66 |
|  | MSP-TS-AURC | 100 | 72.89 | 55.83 | 40.90 | 24.65 | 12.46 | 77.29 |
|  | MaxLogit-pNorm | 100 | **75.24** | **58.76** | **43.98** | **27.27** | **14.78** | **79.00** |

expense of coverage; meanwhile, MaxLogit-pNorm achieves higher coverages for all distribution shifts considered, significantly improving coverage over the MSP baseline.

## 5  CONCLUSION

In this paper, we addressed the problem of selective multiclass classification for deep neural networks with softmax outputs. Specifically, we considered the design of post-hoc confidence estimators that can be computed directly from the unnormalized logits. We performed an extensive benchmark of more than 20 tunable post-hoc methods across 84 ImageNet classifiers, establishing strong baselines for future research. To allow for a fair comparison, we proposed a normalized version of the AURC metric that is insensitive to the classifier accuracy.

Our main conclusions are the following: (1) For 58 (69%) of the models considered, considerable NAURC gains over the MSP can be obtained, in one case achieving a reduction of 0.27 points or about 62%. (2) Our proposed method MaxLogit-pNorm (which does not use a softmax function) emerges as a clear winner, providing the highest gains with exceptional data efficiency, requiring on average less than 1 sample per class of hold-out data for tuning its single hyperparameter. These observations are also confirmed under additional datasets and the gains preserved even under distribution shift. (3) After post-hoc optimization, all models achieve a similar level of confidence estimation performance, even models that have been previously shown to be very poor at this task. In particular, the selective classification performance of any classifier becomes almost entirely determined by its corresponding accuracy, eliminating the seemingly existing tradeoff between these two goals reported in previous work. (4) Selective classification performance itself appears to be robust to distribution shift, in the sense that, although it naturally degrades, this degradation is not larger than what would be expected by the corresponding accuracy drop.

Two questions naturally emerge from our results, which are left as suggestions for future work. Can better performance be attainable with more complex post-hoc methods under limited (or even unlimited) tuning data? What exact properties of a classifier or training regime make it improvable by post-hoc methods? Our investigation suggests that the issue is related to underconfidence, but a complete explanation is still elusive.

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

## A   RELATED WORK

Selective prediction is also known as learning with a reject option (see (Zhang et al., 2023; Hendrickx et al., 2021) and references therein), where the rejector is usually a thresholded confidence estimator. Essentially the same problem is studied under the equivalent terms misclassification detection (Hendrycks & Gimpel, 2016), failure prediction (Corbière et al., 2022; Zhu et al., 2022), and (ordinal) ranking (Moon et al., 2020; Galil et al., 2023). Uncertainty estimation is a more general term that encompasses these tasks (where confidence may be taken as negative uncertainty) as well as other tasks where uncertainty might be useful, such as calibration and out-of-distribution (OOD) detection, among others (Gawlikowski et al., 2022; Abdar et al., 2021). These tasks are generally not aligned: for instance, optimizing for calibration may harm selective classification performance (Ding et al., 2020; Zhu et al., 2022; Galil et al., 2023). Our focus here is on in-distribution selective classification, although we also study robustness to distribution shift.

Interestingly, the same principles of selective classification can be applied to enable efficient inference with model cascades (Lebovitz et al., 2023), although the literature on those topics appears disconnected.

Most approaches to selective classification consider the base model as part of the learning problem (Geifman & El-Yaniv, 2019; Huang et al., 2020; Liu et al., 2019b), which we refer to as training-based approaches. While such an approach has a theoretical appeal, the fact that it requires retraining a model is a significant practical drawback. Alternatively, one may keep the model fixed and only modify or replace the confidence estimator, which is known as a post-hoc approach. Such an approach is practically appealing and perhaps more realistic, as it does not require retraining. Papers that follow this approach typically construct a *meta-model* that feeds on intermediate features of the base model and is trained to predict whether or not the base model is correct on hold-out samples (Corbière et al., 2022; Shen et al., 2022). However, depending on the size of such a meta-model, its training may still be computationally demanding.

A popular tool in the uncertainty literature is the use of ensembles (Lakshminarayanan et al., 2017; Teye et al., 2018; Ayhan & Berens, 2018), of which Monte-Carlo dropout Gal & Ghahramani (2016) is a prominent example. While constructing a confidence estimator from ensemble component outputs may be considered post-hoc if the ensemble is already trained, the fact that multiple inference passes need to be performed significantly increases the computational burden at test time. Moreover, recent work has found evidence that ensembles may not be fundamental for uncertainty but simply better predictive models (Abe et al., 2022; Cattelan & Silva, 2022; Xia & Bouganis, 2022). Thus, we do not consider ensembles here.

In this work we focus on simple post-hoc confidence estimators for softmax networks that can be directly computed from the logits. The earliest example of such a post-hoc method used for selective classification in a real-world application seems to be the use of LogitsMargin in (Le Cun et al., 1990). While potentially suboptimal, such methods are extremely simple to apply on top of any trained classifier and should be natural choice to try before any more complex technique. In fact, it is not entirely obvious how a training-based approach should be compared to a post-hoc method. For instance, Feng et al. (2023) has found that, for some state-of-the-art training-based approaches to selective classification, *after* the main classifier has been trained with the corresponding technique, better selective classification performance can be obtained by discarding the auxiliary output providing confidence values and simply use the conventional MSP as the confidence estimator. Thus, in this sense, the MSP can be seen as a strong baseline.

Post-hoc methods have been widely considered in the context of calibration, among which the most popular approach is temperature scaling (TS). Applying TS to improve calibration (of the MSP confidence estimator) was originally proposed in (Guo et al., 2017) based on the negative log-likelihood. Optimizing TS for other metrics has been explored in (Mukhoti et al., 2020; Karandikar et al., 2021; Clarté et al., 2023) for calibration and in (Liang et al., 2023) for OOD detection, but had not been proposed for selective classification. A generalization of TS is adaptive TS (ATS) (Balanya et al., 2023), which uses an input-dependent temperature based on logits. The post-hoc methods we consider here can be seen as a special case of ATS, as logit norms may be seen as an input-dependent temperature; however Balanya et al. (2023) investigate a different temperature function and focuses on calibration. (For more discussion on calibration methods, please see Appendix H.) Other logit-based confidence estimators proposed for calibration and OOD detection include (Liu et al., 2020; Tomani et al., 2022; Rahimi et al., 2022; Neumann et al., 2018; Gonsior et al., 2022).

Normalizing the logits with the $L_2$ norm before applying the softmax function was used in (Kornblith et al., 2021) and later proposed and studied in (Wei et al., 2022) as a training technique (combined with TS) to improve OOD detection and calibration. A variation where the logits are normalized to unit variance was proposed in (Jiang et al., 2023) to accelerate training.

Benchmarking of models in their performance at selective classification/misclassification detection has been done in (Galil et al., 2023; Ding et al., 2020), however these works mostly consider the MSP as the confidence estimator. In the context of calibration, Wang et al. (2021) and Ashukha et al. (2020) have argued that models should be compared after simple post-hoc optimizations, since models that appear worse than others can sometimes easily be improved by methods such as TS. Here we advocate and provide further evidence for this approach in the context of selective classification.

## B   ON THE DOCTOR METHOD

The paper by (Granese et al., 2021) introduces a selection mechanism named DOCTOR, which actually refers to two distinct methods, $D_\alpha$ and $D_\beta$, in two possible scenarios, Total Black Box and Partial Black Box. Only the former scenario corresponds to post-hoc estimators and, in this case, the two methods are equivalent to NegativeGini and MSP, respectively.

To see this, first consider the definition of $D_\alpha$: a sample $x$ is rejected if $1 - \hat{g}(x) > \gamma \hat{g}(x)$, where

$$1 - \hat{g}(x) = \sum_{k \in \mathcal{Y}} (\sigma(\mathbf{z}))_k (1 - (\sigma(\mathbf{z}))_k) = 1 - \sum_{k \in \mathcal{Y}} (\sigma(\mathbf{z}))_k^2 = 1 - \|\sigma(\mathbf{z})\|_2^2$$

is exactly the Gini index of diversity applied to the softmax outputs. Thus, a sample $x$ is accepted if $1 - \hat{g}(x) \le \gamma \hat{g}(x) \iff (1+\gamma)\hat{g}(x) \ge 1 \iff \hat{g}(x) \ge 1/(1+\gamma) \iff \hat{g}(x) - 1 \ge 1/(1+\gamma) - 1$. Therefore, the method is equivalent to the confidence estimator $g(x) = \hat{g}(x) - 1 = \|\sigma(\mathbf{z})\|^2 - 1$, with $t = 1/(1+\gamma) - 1$ as the selection threshold.

Now, consider the definition of $D_\beta$: a sample $x$ is rejected if $\hat{P}_e(x) > \gamma(1 - \hat{P}_e(x))$, where $\hat{P}_e(x) = 1 - (\sigma(\mathbf{z}))_{\hat{y}}$ and $\hat{y} = \arg\max_{k \in \mathcal{Y}}(\sigma(\mathbf{z}))_k$, i.e., $\hat{P}_e(x) = 1 - \text{MSP}(\mathbf{z})$. Thus, a sample $x$ is accepted if $\hat{P}_e(x) \le \gamma(1 - \hat{P}_e(x)) \iff (1+\gamma)\hat{P}_e(x) \le \gamma \iff \hat{P}_e(x) \le \gamma/(1+\gamma) \iff \text{MSP}(\mathbf{z}) \ge 1 - \gamma/(1+\gamma) = 1/(1+\gamma)$. Therefore, the method is equivalent to the confidence estimator $g(x) = \text{MSP}(\mathbf{z})$, with $t = 1/(1+\gamma)$ as the selection threshold.

Given the above results, one may wonder why the results in (Granese et al., 2021) show different performance values for $D_\beta$ and MSP (softmax response), as shown, for instance, in Table 1 in Granese et al. (2021). We suspect this discrepancy is due to numerical imprecision in the computation of the ROC curve for a limited number of threshold values, as the authors themselves point out on their Appendix C.3, combined with the fact that $D_\beta$ and MSP in (Granese et al., 2021) use different parametrizations for the threshold values. In contrast, we use the implementation from the scikit-learn library (adapting it as necessary for the RC curve), which considers every possible threshold for the confidence values given and so is immune to this kind of imprecision.

## C   ON LOGIT NORMALIZATION

**Logit normalization during training.** Wei et al. (2022) argued that, as training progresses, a model may tend to become overconfident on correctly classified training samples by increasing $\|\mathbf{z}\|_2$. This is due to the fact that the predicted class depends only on $\tilde{\mathbf{z}} = \mathbf{z}/\|\mathbf{z}\|_2$, but the training loss on correctly classified training samples can still be decreased by increasing $\|\mathbf{z}\|_2$ while keeping $\tilde{\mathbf{z}}$ fixed. Thus, the model would become overconfident on those samples, since increasing $\|\mathbf{z}\|_2$ also increases the confidence (as measured by MSP) of the predicted class. This overconfidence phenomenon was confirmed experimentally in (Wei et al., 2022) by observing that the average magnitude of logits (and therefore also their average 2-norm) tends to increase during training. For this reason, Wei et al. (2022) proposed logit 2-norm normalization during training, as a way to mitigate overconfidence. However, during inference, they still used the raw MSP as confidence estimator, without any normalization.

**Post-training logit normalization.** Here, we propose to use logit $p$-norm normalization as a post-hoc method and we intuitively expected it to have a similar effect in combating overconfidence. (Note that the argument in (Wei et al., 2022) holds unchanged for any $p$, as nothing in their analysis requires $p = 2$.) Our initial hypothesis was the following: if the model has become too overconfident (through high logit norm) on certain input regions, then—since overconfidence is a form of (loss) overfitting—there would be an increased chance that the model will produce incorrect predictions on the test set along these input regions. Thus, high logit norm on the test set would indicate regions of higher inaccuracy, so that, by applying logit normalization, we would be penalizing likely inaccurate predictions, improving selective classification performance. However, this hypothesis was *disproved* by the experimental results in Appendix F, which show that overconfidence is *not* a problem for selective classification, but *underconfidence* may be.

**Combating underconfidence with temperature scaling.** If a model is underconfident on a set of samples, with low logit norm and an MSP value smaller than its expected accuracy on these samples, then the MSP may not provide a good estimate of confidence. In these cases, the LogitsMargin (the margin between the highest and the second highest logit) may be a better confidence estimator.

Alternatively, one may use MSP-TS with a low temperature, which approximates LogitsMargin, as can be easily seen below. Let $\mathbf{z} = (z_1, \ldots, z_C)$, with $z_1 > \ldots > z_C$. Then

$$\text{MSP}(\mathbf{z}/T) = \frac{e^{z_1/T}}{\sum_j e^{z_j/T}} = \frac{1}{1 + e^{(z_2 - z_1)/T} + \sum_{j>2} e^{(z_j - z_1)/T}} \tag{11}$$

$$= \frac{1}{1 + e^{-(z_1 - z_2)/T} \left(1 + \sum_{j>2} e^{-(z_2 - z_j)/T}\right)} \approx \frac{1}{1 + e^{-(z_1 - z_2)/T}} \tag{12}$$

for small $T > 0$. Note that a strictly increasing transformation does not change the ordering of confidence values and thus maintains selective classification performance. This helps explain why TS (with $T < 1$) can improve selective classification performance, as already observed in (Galil et al., 2023).

**Logit $p$-norm normalization as temperature scaling.** To shed light on why post-hoc logit $p$-norm normalization (with a general $p$) may be helpful, we can show that it is closely related to MSP-TS. Let $g_p(\mathbf{z}) = z_1/\|\mathbf{z}\|_p$ denote MaxLogit-pNorm. Then

$$\text{MSP}(\mathbf{z}/T) = \left(\frac{e^{z_1}}{\left(\sum_j e^{z_j/T}\right)^T}\right)^{1/T} = \left(\frac{e^{z_1}}{\|e^{\mathbf{z}}\|_{1/T}}\right)^{1/T} = \left(g_{1/T}(e^{\mathbf{z}})\right)^{1/T}. \tag{13}$$

Thus, MSP-TS is equivalent to MaxLogit-pNorm with $p = 1/T$ applied to the transformed logit vector $\exp(\mathbf{z})$. This helps explain why a general $p$-norm normalization is useful, as it is closely related to TS, emphasizing the largest components of the logit vector. This also implies that any benefits of MaxLogit-pNorm over MSP-TS stem from *not* applying the exponential transformation of logits. Why this happens to be useful is still elusive at this point.

Nevertheless, it should be clear that, despite their similarities, logit L2 normalization during training and post-hoc logit $p$-norm normalization are different techniques applied to different problems and with different behavior. Moreover, even if logit normalization during training turns out to be beneficial to selective classification (an evaluation that is however outside the scope of this work), it should be emphasized that post-hoc optimization can be easily applied on top of any trained model without requiring modifications to its training regime.

## D   MORE DETAILS AND RESULTS ON THE EXPERIMENTS ON IMAGENET

### D.1   HYPERPARAMETER OPTIMIZATION OF POST-HOC METHODS

For not being differentiable, the NAURC metric demands a zero-order optimization. For this work, the optimizations of $p$ and $T$ were conducted via grid-search. Note that, as $p$ approaches infinity, $\|\mathbf{z}\|_p \to \max(|\mathbf{z}|)$. Indeed, it tends to converge reasonable quickly. Thus, the grid search on $p$ can be made only for small $p$. In our experiments, we noticed that it suffices to evaluate a few values of $p$, such as the integers between 0 and 10, where the 0-norm is taken here to mean the sum of all nonzero values of the vector. The temperature values were taken from the range between 0.01 and 3, with a step size of 0.01, as this showed to be sufficient for achieving the optimal temperature for selective classification (in general between 0 and 1).

### D.2   AUROC RESULTS

Table 4 shows the AUROC results for all methods for an EfficientNetV2-XL on ImageNet, while Table 5 shows the same but for a VGG-16. As it can be seen, the results are consistent with the ones for NAURC presented in Section 4.

### D.3   RC CURVES

In Figure 5 the RC curves of selected post-hoc methods applied to a few representative models are shown.

Table 4: AUROC[x100] for post-hoc methods for an EfficientNet-V2-XL for ImageNet

| | Logit Transformation | | | |
|---|---|---|---|---|
| Confidence Estimation | Raw | TS-NLL | TS-AUROC | pNorm |
| MSP | 0.7732 | 0.8109 | 0.8587 | 0.8708 |
| SoftmaxMargin | 0.7990 | 0.8246 | 0.8590 | 0.8718 |
| MaxLogit | 0.6347 | 0.6347 | 0.6347 | 0.8741 |
| LogitsMargin | 0.8604 | 0.8604 | 0.8604 | 0.8701 |
| NegativeEntropy | 0.6890 | 0.7710 | 0.7538 | 0.8238 |
| NegativeGini | 0.7669 | 0.8101 | 0.8588 | 0.8711 |

Table 5: AUROC[x100] for post-hoc methods for VGG16 for ImageNet

| | Logit Transformation | | | |
|---|---|---|---|---|
| Confidence Estimation | Raw | TS-NLL | TS-AUROC | pNorm |
| MSP | 0.8661 | 0.8652 | 0.8662 | 0.8662 |
| SoftmaxMargin | 0.8603 | 0.8610 | 0.8617 | 0.8617 |
| MaxLogit | 0.7884 | 0.7884 | 0.7884 | 0.8557 |
| LogitsMargin | 0.8478 | 0.8478 | 0.8478 | 0.8478 |
| NegativeEntropy | 0.8556 | 0.8492 | 0.8659 | 0.8659 |
| NegativeGini | 0.8645 | 0.8619 | 0.8661 | 0.8661 |

### D.4    EFFECT OF $\epsilon$

Figure 6 shows the results (in APG metric) for all methods when $p$ is optimized. As can be seen, MaxLogit-pNorm is dominant for all $\epsilon > 0$, indicating that, provided the MSP fallback described in Section 3.2.2 is enabled, it outperforms the other methods.

### D.5    SAC RESULTS

The AURC and NAURC metrics are summarizations of the operation points for all possible coverages. However, once a desired risk is determined as a constraint, a common practical scenario in which selective classification is applied Geifman & El-Yaniv (2017), the SAC metric can be more useful. The SACs for accuracy 98% for the baseline and the optimized confidence estimator are presented in Figure 7. Note that, as the NAURC and the AURC, the curve tends to be monotonic after optimization, as it is the ideal case. Moreover, note that some models fail to obtain this accuracy (coverage $\approx 0$) using the baseline. This problem is mitigated with the post-hoc optimization.

## E    EXPERIMENTS ON ADDITIONAL DATASETS

### E.1    EXPERIMENTS ON OXFORD-IIIT PET

The hold-out set for Oxford-IIIT Pet, consisting of 500 samples, was taken from the training set before training. The model used was an EfficientNet-V2-XL pretrained on ImageNet from Wightman (2019). It was fine-tuned on Oxford-IIIT Pet (Parkhi et al., 2012). The training was conducted for 100 epochs with Cross Entropy Loss, using a SGD optimizer with initial learning rate of 0.1 and a Cosine Annealing learning rate schedule with period 100. Moreover, a weight decay of 0.0005 and a Nesterov's momentum of 0.9 were used. Data transformations were applied, specifically standardization, random crop (for size 224x224) and random horizontal flip.

Figure 8 shows the RC curves for some selected methods for the EfficientNet-V2-XL. As can be seen, considerable gains are obtained with the optimization of $p$, especially in the low-risk region.

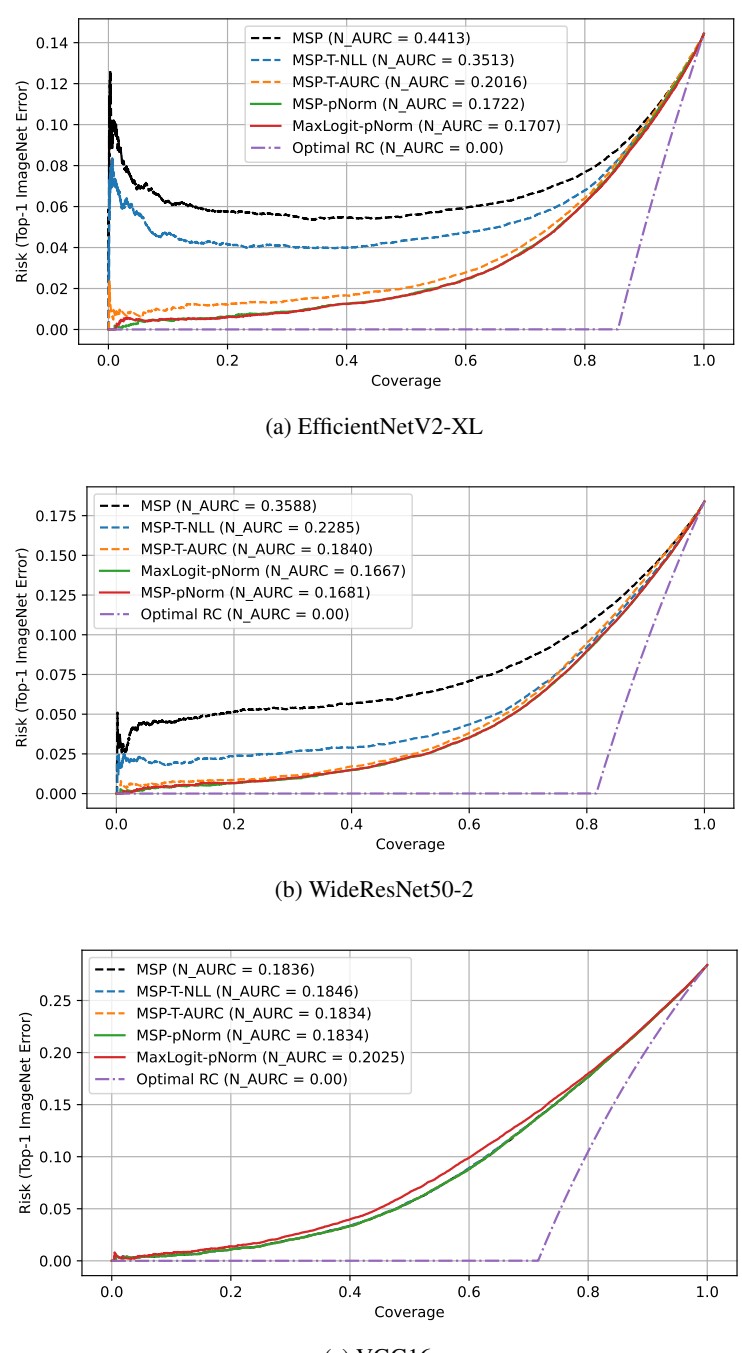

Figure 5: RC curves for selected post-hoc methods applied to ImageNet classifiers.

### E.2 EXPERIMENTS ON CIFAR-100

The hold-out set for CIFAR-100, consisting of 5000 samples, was taken from the training set before training. The model used was forked from `github.com/kuangliu/pytorch-cifar`, and adapted for CIFAR-100 (Krizhevsky, 2009). It was trained for 200 epochs with Cross Entropy Loss, using a SGD optimizer with initial learning rate of 0.1 and a Cosine Annealing learning rate schedule with period 200. Moreover, a weight decay of 0.0005 and a Nesterov's momentum of 0.9 were used.

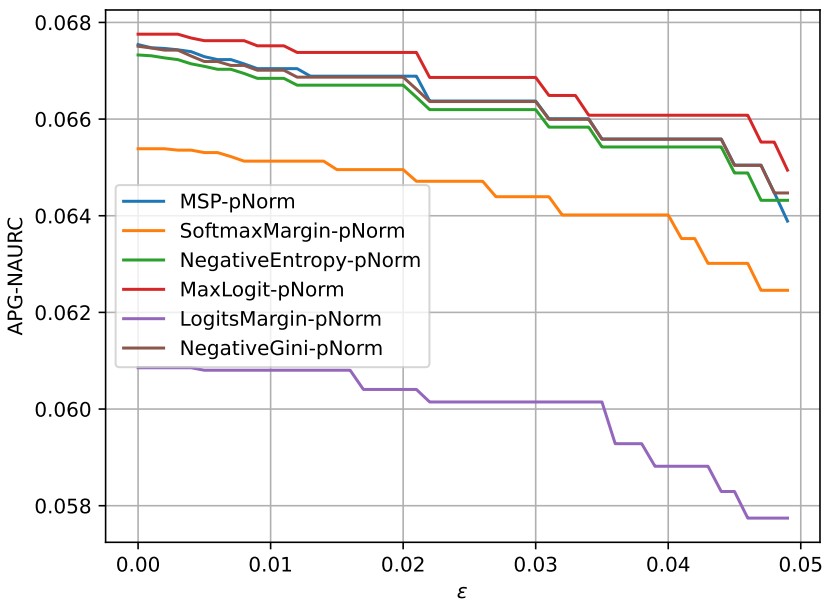

Figure 6: APG in terms of $\epsilon$

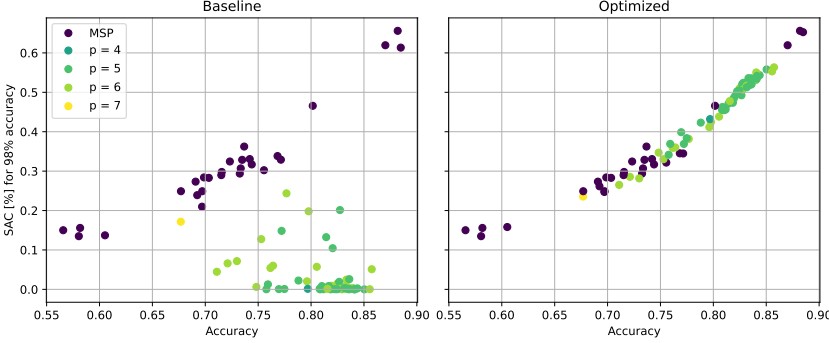

Figure 7: SAC for accuracy 98% of 84 ImageNet classifiers with respect to their accuracy, before and after post-hoc optimization with MaxLogit-pNorm. The legend shows the optimal value of p for each model, where MSP indicates MSP fallback (no significant positive gain)

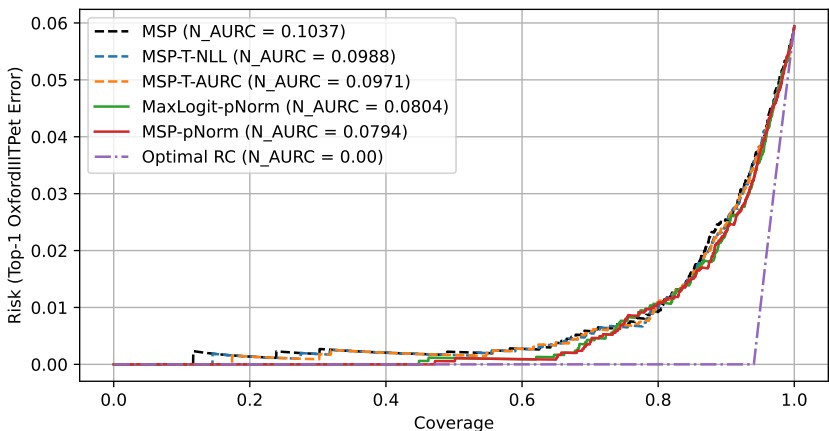

Figure 8: RC curves for a EfficientNet-V2-XL for Oxford-IIIT Pet

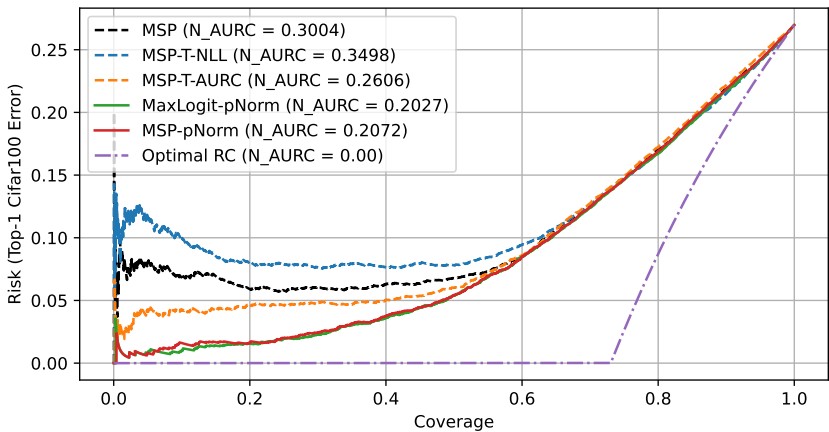

Figure 9: RC curves for a VGG19 for CIFAR-100

Data transformations were applied, specifically standardization, random crop (for size 32x32 with padding 4) and random horizontal flip.

Figure 9 shows the RC curves for some selected methods for a VGG19. As it can be seen, the results follow the same pattern of the ones observed for ImageNet, with MaxLogit-pNorm achieving the best results.

## F  DATA EFFICIENCY

In this section, we empirically investigate the *data efficiency* of tunable post-hoc methods, which refers to their ability to learn and generalize from limited data. As is well-known from machine learning theory and practice, the more we evaluate the empirical risk to tune a parameter, the more we are prone to overfitting, which is aggravated as the size of the dataset used for tuning decreases. Thus, a method that require less hyperparameter tuning tends to be more data efficient, i.e., to achieve its optimal performance with less tuning data. We intuitively expect this to be the case for MaxLogit-pNorm, which only requires evaluating a few values of $p$, compared to any method based on the softmax function, which requires tuning a temperature parameter.

As mentioned in Section 4, the experiments conducted in ImageNet used a hold-out tuning dataset of 5000 images randomly sampled from the available ImageNet validation dataset, resulting in 45000 images reserved for the test phase. To evaluate data efficiency, the optimization process was executed multiple times, utilizing different fractions of the tuning set while keeping the test set fixed at 45000

samples. Consequently, two distinct types of random splits were implemented: the first involved dividing the available dataset into hold-out and test sets (with 5000 and 45000 images, respectively), while the second involved sampling the tuning set as a subset of the hold-out set. To ensure the findings are statistically significant, both of these random split procedures were repeated five times each, culminating in a total of 25 realizations for each size of the tuning set considered.

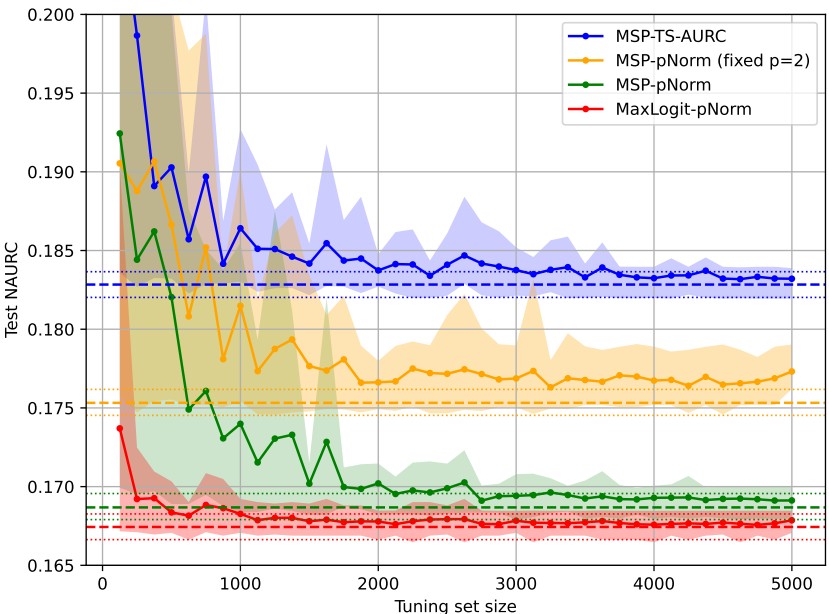

Figure 10: Data Efficiency: Average NAURC variation with number of hold-out samples used, for a ResNet50 on ImageNet. Dashed lines represent the optimal NAURC for each method, i.e., the achieved value when the optimization is made directly on the test set. Filled regions for each curve correspond to percentiles 10 and 90, while dotted lines represent the same but for the optimal value (optimized on test set directly). For comparison, the MSP attains an average NAURC of 0.3209.

Figure 10 displays the outcomes of these studies for a ResNet50 trained on ImageNet. As observed, MaxLogit-pNorm exhibits outstanding data efficiency, while methods that require temperature optimization achieve lower efficiency.

Additionally, the data efficiency experiment was conducted on the VGG19 model for CIFAR-100. Indeed, the same conclusions hold about the high efficiency of MaxLogit-pNorm.

## G  ABLATION STUDY ON THE CHOICE OF $p$

A natural question regarding $p$-norm normalization (with a general $p$) is whether it can provide any benefits beyond the default $p = 2$ used by Wei et al. (2022). Table 6 shows the APG-NAURC results for the 84 ImageNet classifiers when different values of $p$ are kept fixed and when $p$ is optimized for each model (tunable).

As can be seen, there is a significant benefit of using a larger $p$ (especially a tunable one) compared to just using $p = 2$, especially for MaxLogit-pNorm. The statistical significance of these gains (which must be evaluated separately for each model) is discussed in Appendix I. Note that, differently from MaxLogit-pNorm, MSP-pNorm requires temperature optimization. This additional tuning is detrimental to data efficiency, as shown in Appendices F and I.

## H  COMPARISON WITH OTHER TUNABLE METHODS

In Section 4.1 we compared several logit-based confidence estimators obtained by combining a parameterless confidence estimator with a tunable logit transformation, specifically, TS and $p$-norm

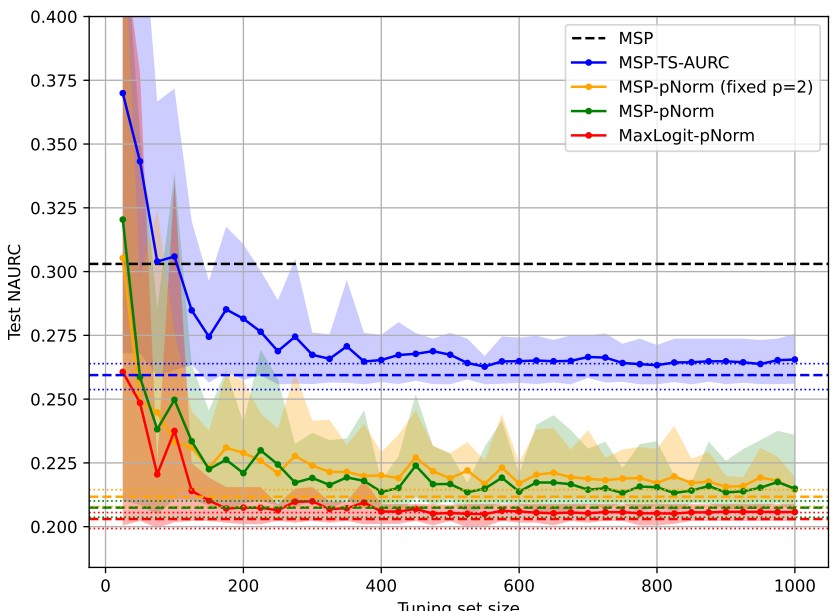

Figure 11: Data Efficiency: Average NAURC variation with number of hold-out samples used, for a VGG19 for CIFAR100. Dashed lines represent the optimal NAURC for each method, i.e., the achieved value when the optimization is made directly on the test set. Filled regions for each curve correspond to percentiles 10 and 90, while dotted lines represent the same but for the optimal value (optimized on test set directly).

Table 6: APG-NAURC across 84 ImageNet classifiers, for different values of $p$

|  | Confidence Estimator | |
|---|---|---|
| $p$ | MaxLogit-pNorm | MSP-pNorm |
| 0 | 0.00000 | 0.05776 |
| 1 | 0.00197 | 0.06030 |
| 2 | 0.01471 | 0.06501 |
| 3 | 0.05039 | 0.06714 |
| 4 | 0.06416 | 0.06798 |
| 5 | 0.06778 | 0.06787 |
| 6 | 0.06785 | 0.06741 |
| 7 | 0.06654 | 0.06703 |
| Tunable | 0.06837 | 0.06781 |

normalization. In this section, we consider other previously proposed tunable confidence estimators that do not fit into this framework.

Note that some of these methods were originally proposed seeking calibration and hence its hyperparameters were tuned to optimize the NLL loss (which is usually suboptimal for selective classification). Instead, to make a fair comparison, we optimized all of their parameters using the AURC metric as the objective.

Zhang et al. (2020) proposed *ensemble temperature scaling* (ETS):

$$\text{ETS}(\mathbf{z}) \triangleq w_1 \text{MSP}\left(\frac{\mathbf{z}}{T}\right) + w_2 \text{MSP}(\mathbf{z}) + w_3 \frac{1}{C} \tag{14}$$

where $w_1, w_2, w_3 \in \mathbb{R}^+$ are tunable parameters and $T$ is the temperature previously obtained through the temperature scaling method. The grid for both $w_1$ and $w_2$ was $[0, 1]$ as suggested by the authors, with a step size of 0.01, while the parameter $w_3$ was not considered since the sum of a constant to the

confidence estimator cannot change the ranking between samples, and consequently cannot change the value of selective classification metrics.

Boursinos & Koutsoukos (2022) proposed the following confidence estimator, referred to here as *Boursinos-Koutsoukos* (BK):

$$\text{BK}(\mathbf{z}) \triangleq a\text{MSP}(\mathbf{z}) + b(1 - \max_{k \in \mathcal{Y}: k \neq \hat{y}} \sigma_k(\mathbf{z})) \tag{15}$$

where $a, b \in \mathbb{R}$ are tunable parameters. The grid for both $a$ and $b$ was $[-1, 1]$ as suggested by the authors, with a step size of 0.01, though we note that the optimization never found $a < 0$ (probably due the high value of the MSP as a confidence estimator).

Finally, Balanya et al. (2023) proposed *entropy-based temperature scaling* (HTS):

$$\text{HTS}(\mathbf{z}) \triangleq \text{MSP}\left(\frac{\mathbf{z}}{T_H(\mathbf{z})}\right) \tag{16}$$

where $T_H(\mathbf{z}) = \log\left(1 + \exp(b + w\log\bar{H}(\mathbf{z}))\right)$, $\bar{H}(\mathbf{z}) = -(1/C)\sum_{k \in \mathcal{Y}} \sigma_k(\mathbf{z})\log\sigma_k(\mathbf{z})$, and $b, w \in \mathbb{R}$ are tunable parameters. The grids for $b$ and $w$ were, respectively, $[-3, 1]$ and $[-1, 1]$, with a step size of 0.01, and we note that the optimal parameters were always strictly inside the grid.

The results for these post-hoc methods are shown in Table 7. Interestingly, BK, which can be seen as a tunable linear combination of MSP and SoftmaxMargin, is able to outperform both of them, although it still underperforms MSP-TS. On the other hand, ETS, which is a tunable linear combination of MSP and MSP-TS, attains exactly the same performance as MSP-TS. Finally, HTS, which is a generalization of MSP-TS, is able to outperform it, although it still underperforms most methods that use $p$-norm tuning (see Table 2). In particular, MaxLogit-pNorm shows superior performance to all of these methods, while requiring much less hyperparameter tuning.

Table 7: APG-NAURC of additional tunable post-hoc methods across 84 ImageNet classifiers

| Method | APG-NAURC |
|---|---|
| ETS | 0.05776 |
| BK | 0.02504 |
| HTS | 0.06286 |
| MaxLogit-pNorm | **0.06837** |

Methods with a larger number of tunable parameters, such as PTS (Tomani et al., 2022) and HnLTS (Balanya et al., 2023), are only viable with a differentiable loss. As these methods are proposed for calibration, the NLL loss is used; however, as previous works have shown that this does not always improve and sometimes even harm selective classification (Zhu et al., 2022; Galil et al., 2023), these methods were not considered in our work. The investigation of alternative methods for optimizing selective classification (such as proposing differentiable losses or more efficient zero-order methods) is left as a suggestion for future work. In any case, note that using a large number of hyperparameters is likely to harm data efficiency.

We also evaluated additional parameterless confidence estimators proposed for selective classification (Hasan et al., 2023), such as LDAM (He et al., 2011) and the method in (Leon-Malpartida et al., 2018), both in their raw form and with TS/pNorm optimization, but none of these methods showed any gain over the MSP. Note that the Gini index, sometimes proposed as a post-hoc method (Hasan et al., 2023) (and also known as DOCTOR's $D_\alpha$ method (Granese et al., 2021)) has already been covered in Section 2.2.

## I   STATISTICAL SIGNIFICANCE OF NAURC GAINS

In order to evaluate the statistical significance of the NAURC gains obtained for different models, we repeat each experiment (including post-hoc optimization) for different random splits of the available dataset into tuning and test sets (similarly to the analysis conducted in Appendix F), for each of the 84 ImageNet classifiers considered. Figure 12 considers 5 random splits of 5000 samples for the tuning set, while Figure 13 considers 10 random splits of 1000 samples, with the remaining samples (out of

50000) being left for the test set. As can be seen, the results obtained are consistent with the ones observed previously. In particular, MaxLogit-pNorm provides a statistically significant improvement over MSP-TS and MSP-pNorm with fixed $p = 2$, and also outperforms MSP-pNorm (with tunable $p$) when the tuning set is reduced. Moreover, we can see that MaxLogit-pNorm is the most stable among these methods in terms of variance of gains.

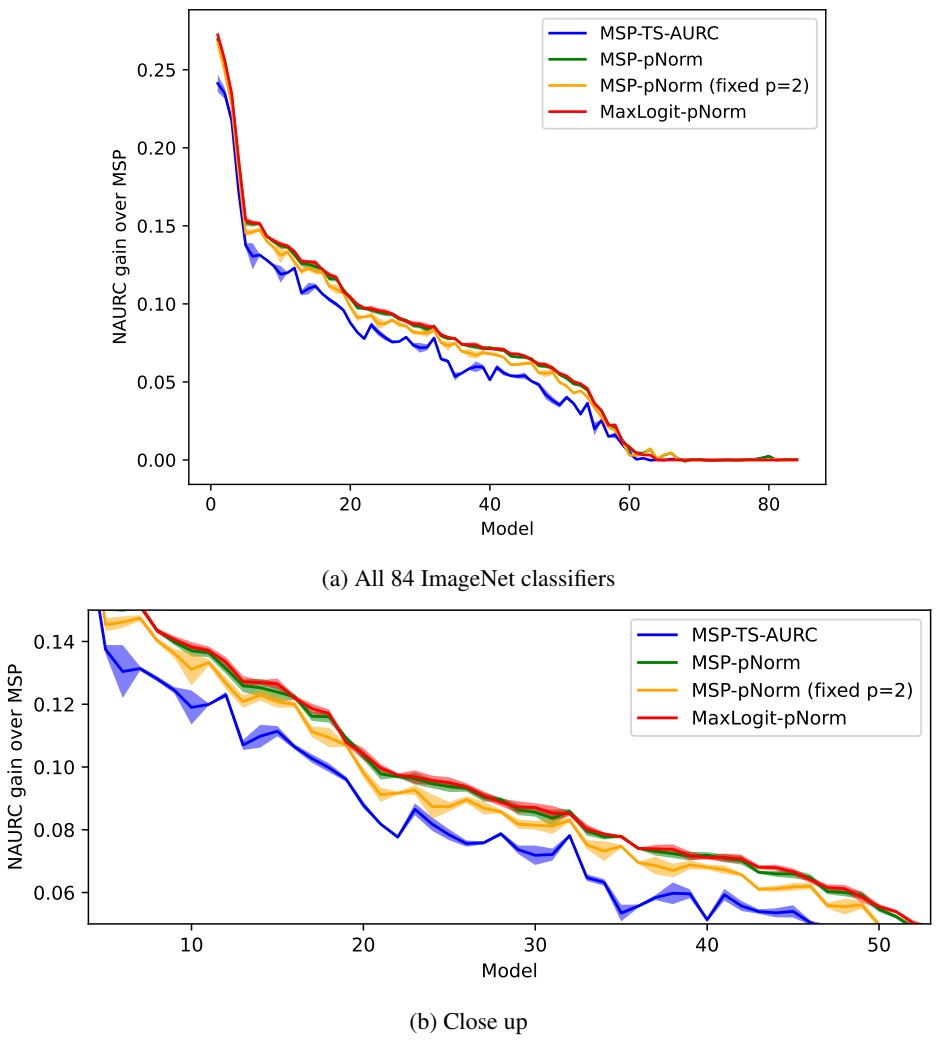

(a) All 84 ImageNet classifiers

(b) Close up

Figure 12: NAURC gains for post-hoc methods (with MSP fallback) using a tuning set with 5000 samples. Lines indicate the average of 5 random splits, and the filled regions indicate ±1 standard deviation.

## J WHEN IS POST-HOC OPTIMIZATION BENEFICIAL?

In this section we investigate in which circumstances post-hoc optimization yields significant gains. Figure 14 shows histograms of confidence values for two representative examples of non-improvable and improvable models, with the latter one shown before and after post-hoc optimization. Figure 15 shows the NAURC gain over MSP versus the proportion of samples with high MSP for each classifier. As can be seen, highly confident models tend to have a good MSP confidence estimator, while less confident models tend to have a poor confidence estimator that is easily improvable by post-hoc methods—after which the resulting confidence estimator becomes concentrated on high values.

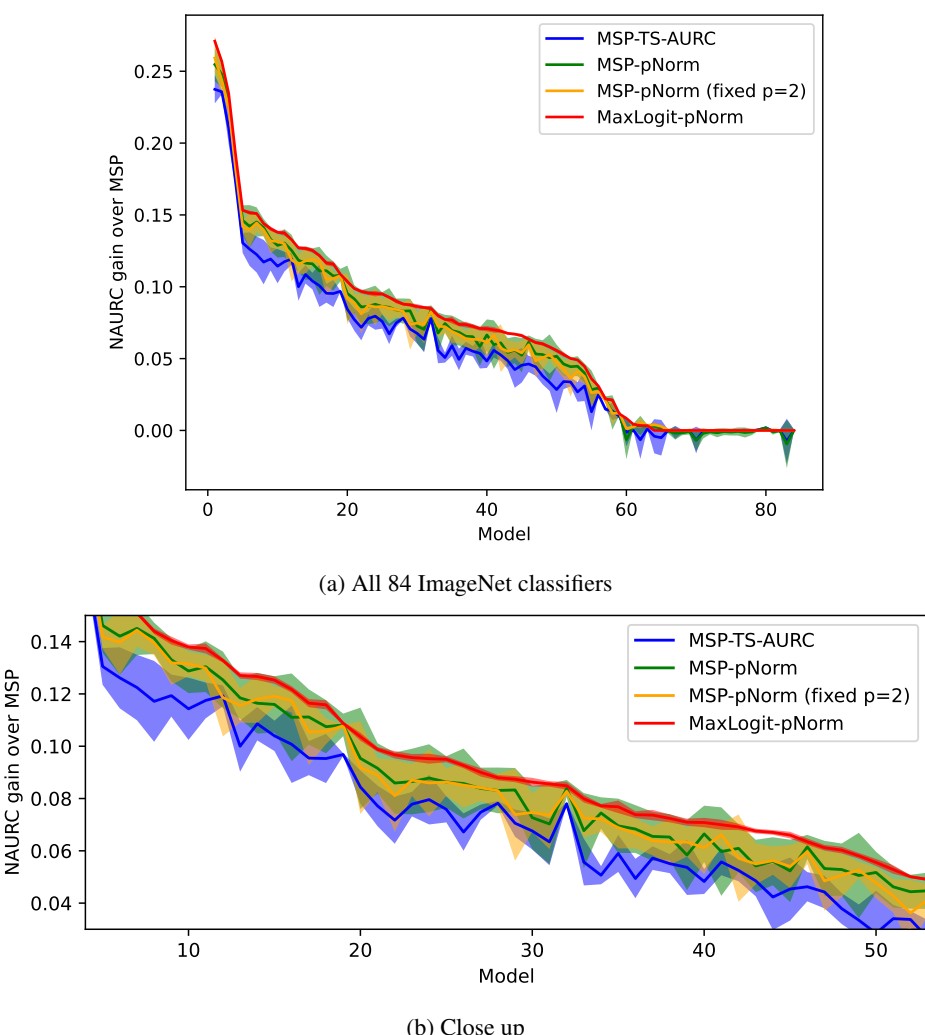

(a) All 84 ImageNet classifiers

(b) Close up

Figure 13: NAURC gains for post-hoc methods (with MSP fallback) using a tuning set with 1000 samples. Lines indicate the average of 10 random splits, and the filled regions indicate ±1 standard deviation.

## K  MORE RESULTS ON DISTRIBUTION SHIFT

Figure 16 provides further insights into our conclusion regarding the strong correlation between NAURC and accuracy, even in the presence of distribution shifts. In this section, we extend our analysis beyond the findings outlined in Section 4.3 presenting the NAURC scores for ResNet50, WideResNet50-2, AlexNet, and ConvNext-Large models tested on ImageNet-C. These additional results consistently reinforce the conclusions drawn earlier.

## L  FULL RESULTS ON IMAGENET

Table 8 presents all the NAURC results for the most relevant methods for all the evaluated models on ImageNet, while Table 9 shows the corresponding AURC results and Table 10 the corresponding AUROC results.

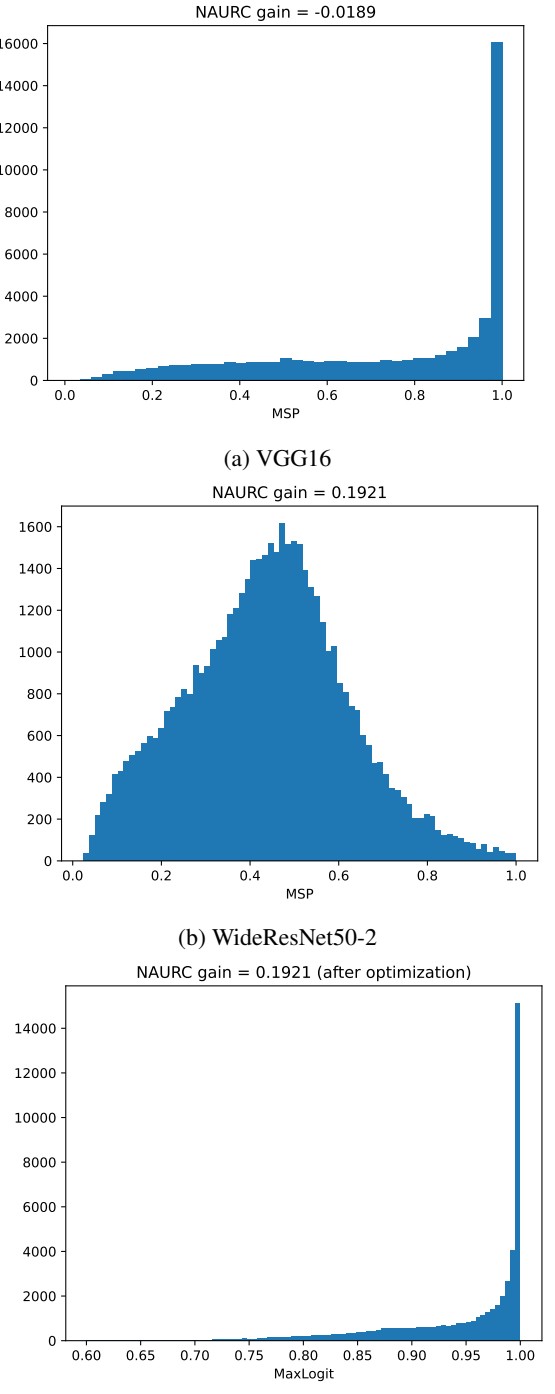

(a) VGG16

(b) WideResNet50-2

(c) WideResNet50-2 after MaxLogit-pNorm optimization

Figure 14: Histograms of confidence values for VGG16 MSP, WideResNet50-2 MSP and WideResNet50-2 MaxLogit-pNorm on ImageNet.

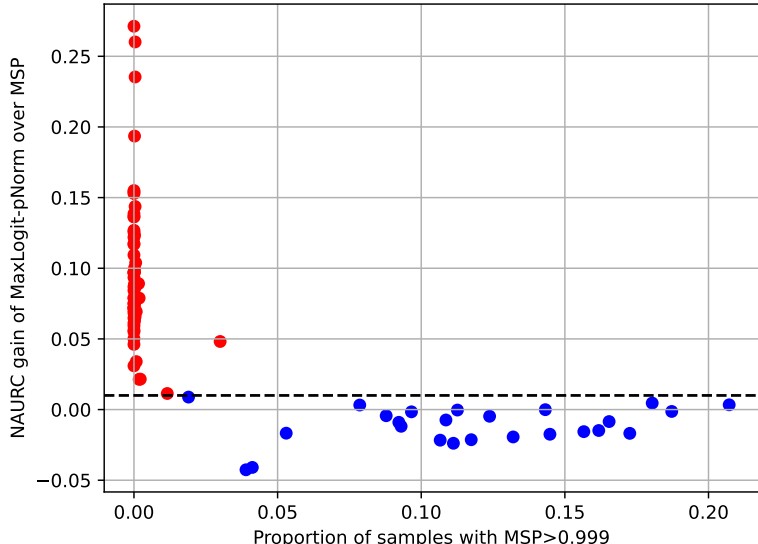

Figure 15: NAURC gain versus the proportion of samples with MSP > 0.999.

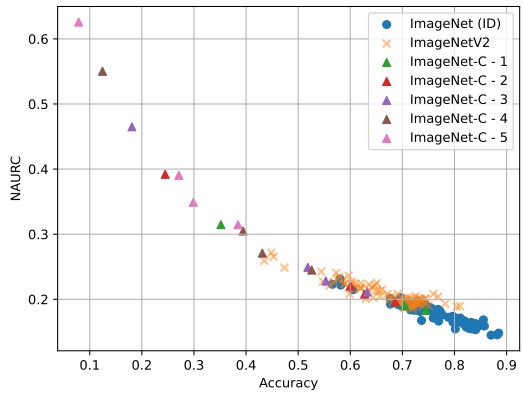

(a) NAURC versus accuracy for the best confidence estimator

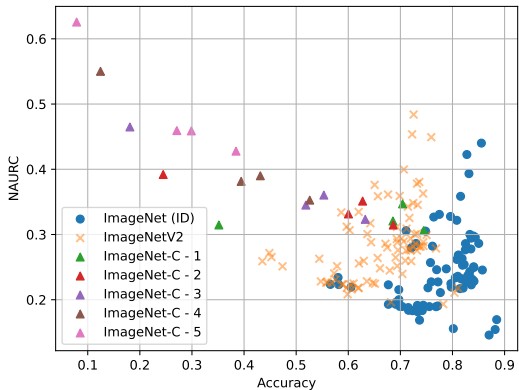

(b) NAURC versus accuracy for the MSP baseline

Figure 16: NAURC under distribution shift for all models considered for ImageNet

Table 8: NAURC for all models evaluated on ImageNet

| Model | Accuracy[%] | Method | | | | | |
| --- | --- | --- | --- | --- | --- | --- | --- |
| | | MSP | MSP-TS-NLL | MSP-TS-AURC | LogitsMargin | MSP-pNorm - $p^*$ | MaxLogit-pNorm - $p^*$ |
| alexnet | 0.5657 | 0.2234 | 0.2248 | 0.2232 | 0.2604 | 0.2232 - 0 | 0.2401 - 0 |
| convnext_base | 0.8402 | 0.2977 | 0.2293 | 0.1795 | 0.1784 | 0.1613 - 4 | 0.1611 - 5 |
| convnext_large | 0.8438 | 0.2956 | 0.2413 | 0.1772 | 0.1728 | 0.1582 - 5 | 0.1574 - 5 |
| convnext_small | 0.8360 | 0.2944 | 0.2249 | 0.1733 | 0.1732 | 0.1580 - 3 | 0.1558 - 5 |
| convnext_tiny | 0.8249 | 0.2860 | 0.2190 | 0.1832 | 0.1810 | 0.1605 - 5 | 0.1592 - 6 |
| densenet121 | 0.7439 | 0.1915 | 0.1926 | 0.1915 | 0.2095 | 0.1914 - 0 | 0.1989 - 0 |
| densenet161 | 0.7715 | 0.1904 | 0.1943 | 0.1903 | 0.2036 | 0.1831 - 3 | 0.1870 - 7 |
| densenet169 | 0.7555 | 0.1895 | 0.1923 | 0.1894 | 0.2046 | 0.1862 - 1 | 0.1907 - 7 |
| densenet201 | 0.7683 | 0.1894 | 0.1911 | 0.1887 | 0.2024 | 0.1844 - 3 | 0.1894 - 7 |
| efficientnet_b0 | 0.7768 | 0.2110 | 0.1962 | 0.1965 | 0.2004 | 0.1761 - 4 | 0.1764 - 6 |
| efficientnet_b1 | 0.7977 | 0.2194 | 0.1884 | 0.1819 | 0.1891 | 0.1745 - 4 | 0.1733 - 6 |
| efficientnet_b2 | 0.8054 | 0.2336 | 0.2062 | 0.1868 | 0.1929 | 0.1714 - 5 | 0.1705 - 6 |
| efficientnet_b3 | 0.8193 | 0.2529 | 0.2066 | 0.1810 | 0.1794 | 0.1651 - 5 | 0.1637 - 6 |
| efficientnet_b4 | 0.8333 | 0.3001 | 0.2147 | 0.1750 | 0.1763 | 0.1692 - 3 | 0.1660 - 7 |
| efficientnet_b5 | 0.8334 | 0.2350 | 0.1977 | 0.1704 | 0.1736 | 0.1574 - 4 | 0.1563 - 6 |
| efficientnet_b6 | 0.8389 | 0.2304 | 0.1930 | 0.1735 | 0.1705 | 0.1735 - 0 | 0.1559 - 0 |
| efficientnet_b7 | 0.8406 | 0.2505 | 0.2031 | 0.1639 | 0.1664 | 0.1550 - 3 | 0.1532 - 6 |
| efficientnet_v2_l | 0.8574 | 0.2456 | 0.2061 | 0.1730 | 0.1734 | 0.1597 - 5 | 0.1590 - 6 |
| efficientnet_v2_m | 0.8504 | 0.2863 | 0.2220 | 0.1768 | 0.1757 | 0.1646 - 3 | 0.1606 - 5 |
| efficientnet_v2_s | 0.8413 | 0.2284 | 0.1918 | 0.1667 | 0.1698 | 0.1563 - 4 | 0.1556 - 5 |
| googlenet | 0.6970 | 0.2154 | 0.2068 | 0.2052 | 0.2279 | 0.2024 - 3 | 0.2040 - 6 |
| inception_v3 | 0.7722 | 0.2273 | 0.2158 | 0.1973 | 0.2026 | 0.1800 - 4 | 0.1791 - 5 |
| maxvit_t | 0.8362 | 0.2228 | 0.2024 | 0.1737 | 0.1735 | 0.1614 - 5 | 0.1600 - 5 |
| mnasnet0_5 | 0.6769 | 0.2229 | 0.2104 | 0.2086 | 0.2324 | 0.2014 - 3 | 0.2013 - 7 |
| mnasnet0_75 | 0.7110 | 0.3059 | 0.2124 | 0.2081 | 0.2252 | 0.1955 - 3 | 0.1964 - 6 |
| mnasnet1_0 | 0.7349 | 0.1833 | 0.1854 | 0.1834 | 0.2010 | 0.1833 - 0 | 0.1917 - 0 |
| mnasnet1_3 | 0.7640 | 0.3268 | 0.2093 | 0.1960 | 0.2039 | 0.1814 - 4 | 0.1815 - 6 |
| mobilenet_v2 | 0.7211 | 0.2795 | 0.2049 | 0.2020 | 0.2204 | 0.1943 - 4 | 0.1952 - 6 |
| mobilenet_v3_large | 0.7529 | 0.2183 | 0.1953 | 0.1926 | 0.2099 | 0.1874 - 5 | 0.1873 - 6 |
| mobilenet_v3_small | 0.6770 | 0.1929 | 0.1946 | 0.1926 | 0.2170 | 0.1926 - 0 | 0.2047 - 0 |

| | | | | | | | |
|---|---|---|---|---|---|---|---|
| regnet_x_16gf | 0.8266 | 0.2271 | 0.2008 | 0.1743 | 0.1745 | 0.1619 - 4 | 0.1613 - 5 |
| regnet_x_1_6gf | 0.7962 | 0.3277 | 0.2104 | 0.1899 | 0.1948 | 0.1740 - 4 | 0.1736 - 6 |
| regnet_x_32gf | 0.8294 | 0.2373 | 0.2127 | 0.1719 | 0.1746 | 0.1621 - 3 | 0.1583 - 5 |
| regnet_x_3_2gf | 0.8111 | 0.2679 | 0.2161 | 0.1905 | 0.1928 | 0.1713 - 5 | 0.1718 - 6 |
| regnet_x_400mf | 0.7483 | 0.3052 | 0.2072 | 0.1989 | 0.2111 | 0.1826 - 4 | 0.1836 - 6 |
| regnet_x_800mf | 0.7748 | 0.3308 | 0.2128 | 0.1986 | 0.2067 | 0.1778 - 4 | 0.1779 - 5 |
| regnet_x_8gf | 0.8166 | 0.2320 | 0.2013 | 0.1754 | 0.1778 | 0.1653 - 3 | 0.1626 - 5 |
| regnet_y_128gf | 0.8819 | 0.1543 | 0.1553 | 0.1489 | 0.1525 | 0.1463 - 4 | 0.1455 - 7 |
| regnet_y_16gf | 0.8284 | 0.2786 | 0.2271 | 0.1726 | 0.1700 | 0.1553 - 4 | 0.1551 - 5 |
| regnet_y_1_6gf | 0.8085 | 0.2617 | 0.2113 | 0.1893 | 0.1910 | 0.1684 - 4 | 0.1682 - 5 |
| regnet_y_32gf | 0.8332 | 0.2432 | 0.2038 | 0.1693 | 0.1701 | 0.1555 - 3 | 0.1544 - 5 |
| regnet_y_3_2gf | 0.8189 | 0.2318 | 0.1980 | 0.1811 | 0.1841 | 0.1669 - 6 | 0.1670 - 5 |
| regnet_y_400mf | 0.7578 | 0.2569 | 0.2144 | 0.2047 | 0.2164 | 0.1839 - 4 | 0.1847 - 5 |
| regnet_y_800mf | 0.7881 | 0.2474 | 0.2036 | 0.1909 | 0.1999 | 0.1725 - 4 | 0.1724 - 5 |
| regnet_y_8gf | 0.8273 | 0.2291 | 0.1928 | 0.1703 | 0.1711 | 0.1576 - 5 | 0.1570 - 5 |
| resnet101 | 0.8185 | 0.2632 | 0.2176 | 0.1825 | 0.1832 | 0.1675 - 5 | 0.1658 - 5 |
| resnet152 | 0.8225 | 0.2540 | 0.2081 | 0.1702 | 0.1716 | 0.1584 - 4 | 0.1573 - 5 |
| resnet18 | 0.6971 | 0.2000 | 0.2018 | 0.1994 | 0.2208 | 0.1999 - 1 | 0.2090 - 7 |
| resnet34 | 0.7326 | 0.1913 | 0.1928 | 0.1914 | 0.2107 | 0.1907 - 2 | 0.1956 - 7 |
| resnet50 | 0.8082 | 0.3218 | 0.2109 | 0.1814 | 0.1841 | 0.1679 - 4 | 0.1667 - 5 |
| resnext101_32x8d | 0.8276 | 0.4226 | 0.2542 | 0.1863 | 0.1836 | 0.1633 - 4 | 0.1633 - 6 |
| resnext101_64x4d | 0.8316 | 0.3933 | 0.2332 | 0.1763 | 0.1746 | 0.1609 - 3 | 0.1579 - 6 |
| resnext50_32x4d | 0.8116 | 0.2685 | 0.2206 | 0.1843 | 0.1870 | 0.1703 - 5 | 0.1683 - 5 |
| shufflenet_v2_x0_5 | 0.6052 | 0.2192 | 0.2218 | 0.2180 | 0.2412 | 0.2147 - 4 | 0.2159 - 7 |
| shufflenet_v2_x1_0 | 0.6924 | 0.1965 | 0.2005 | 0.1961 | 0.2107 | 0.1919 - 4 | 0.1918 - 7 |
| shufflenet_v2_x1_5 | 0.7299 | 0.2863 | 0.2122 | 0.2072 | 0.2233 | 0.1963 - 4 | 0.1970 - 6 |
| shufflenet_v2_x2_0 | 0.7616 | 0.2822 | 0.2041 | 0.1938 | 0.2024 | 0.1787 - 4 | 0.1782 - 6 |
| squeezenet1_0 | 0.5804 | 0.2341 | 0.2363 | 0.2318 | 0.2618 | 0.2317 - 0 | 0.2751 - 0 |
| squeezenet1_1 | 0.5817 | 0.2232 | 0.2249 | 0.2220 | 0.2546 | 0.2220 - 0 | 0.2632 - 0 |
| swin_b | 0.8353 | 0.2778 | 0.2420 | 0.1764 | 0.1773 | 0.1650 - 3 | 0.1604 - 5 |
| swin_s | 0.8316 | 0.2329 | 0.2145 | 0.1813 | 0.1808 | 0.1660 - 4 | 0.1645 - 5 |
| swin_t | 0.8143 | 0.2170 | 0.1956 | 0.1801 | 0.1850 | 0.1678 - 4 | 0.1668 - 5 |
| swin_v2_b | 0.8412 | 0.2494 | 0.2219 | 0.1757 | 0.1780 | 0.1633 - 4 | 0.1625 - 5 |
| swin_v2_s | 0.8365 | 0.2317 | 0.2047 | 0.1688 | 0.1702 | 0.1577 - 4 | 0.1565 - 5 |
| swin_v2_t | 0.8204 | 0.2178 | 0.1929 | 0.1747 | 0.1786 | 0.1637 - 4 | 0.1627 - 5 |
| vgg11 | 0.6909 | 0.1931 | 0.1941 | 0.1926 | 0.2164 | 0.1926 - 0 | 0.2147 - 0 |

| | | | | | | |
|---|---|---|---|---|---|---|
| vgg11_bn | 0.7034 | 0.1896 | 0.1909 | 0.1895 | 0.2114 | 0.1895 - 0 | 0.2134 - 0 |
| vgg13 | 0.6988 | 0.1898 | 0.1907 | 0.1895 | 0.2108 | 0.1895 - 0 | 0.2100 - 0 |
| vgg13_bn | 0.7151 | 0.1880 | 0.1895 | 0.1878 | 0.2098 | 0.1877 - 0 | 0.2073 - 0 |
| vgg16 | 0.7156 | 0.1838 | 0.1850 | 0.1836 | 0.2047 | 0.1836 - 0 | 0.2012 - 0 |
| vgg16_bn | 0.7335 | 0.1814 | 0.1833 | 0.1813 | 0.1993 | 0.1813 - 0 | 0.1962 - 0 |
| vgg19 | 0.7232 | 0.1832 | 0.1841 | 0.1831 | 0.2044 | 0.1831 - 0 | 0.1988 - 0 |
| vgg19_bn | 0.7420 | 0.1834 | 0.1851 | 0.1834 | 0.2008 | 0.1834 - 0 | 0.2002 - 0 |
| vit_b_16 | 0.8102 | 0.2359 | 0.2113 | 0.1815 | 0.1831 | 0.1669 - 4 | 0.1657 - 5 |
| vit_b_32 | 0.7592 | 0.2274 | 0.2092 | 0.1903 | 0.1944 | 0.1719 - 4 | 0.1712 - 5 |
| vit_h_14 | 0.8848 | 0.1695 | 0.1652 | 0.1533 | 0.1557 | 0.1483 - 4 | 0.1480 - 6 |
| vit_l_16 | 0.7969 | 0.2246 | 0.2146 | 0.1841 | 0.1869 | 0.1652 - 4 | 0.1654 - 4 |
| vit_l_32 | 0.7696 | 0.2460 | 0.2290 | 0.1932 | 0.1930 | 0.1664 - 4 | 0.1677 - 5 |
| wide_resnet101_2 | 0.8246 | 0.2765 | 0.2255 | 0.1799 | 0.1769 | 0.1606 - 5 | 0.1595 - 5 |
| wide_resnet50_2 | 0.8155 | 0.3585 | 0.2284 | 0.1830 | 0.1852 | 0.1663 - 4 | 0.1654 - 6 |
| efficientnetv2_xl | 0.8555 | 0.4401 | 0.3504 | 0.2056 | 0.1935 | 0.1714 - 4 | 0.1684 - 6 |
| vit_l_16_384 | 0.8702 | 0.1459 | 0.1461 | 0.1451 | 0.1526 | 0.1450 - 0 | 0.1502 - 0 |
| vit_b_16_sam | 0.8016 | 0.1557 | 0.1555 | 0.1546 | 0.1614 | 0.1545 - 0 | 0.1560 - 0 |
| vit_b_32_sam | 0.7368 | 0.1687 | 0.1682 | 0.1676 | 0.1789 | 0.1676 - 0 | 0.1693 - 0 |

Table 9: AURC for all models evaluated on ImageNet

| Model | Accuracy[%] | Method | | | | | |
|---|---|---|---|---|---|---|---|
| | | MSP | MSP-TS-NLL | MSP-TS-AURC | LogitsMargin | MSP-pNorm - $p^*$ | MaxLogit-pNorm - $p^*$ |
| alexnet | 0.5657 | 0.1840 | 0.1845 | 0.1840 | 0.1959 | 0.1839 - 0 | 0.1894 - 0 |
| convnext_base | 0.8402 | 0.0570 | 0.0470 | 0.0398 | 0.0396 | 0.0371 - 4 | 0.0370 - 5 |
| convnext_large | 0.8438 | 0.0553 | 0.0475 | 0.0383 | 0.0377 | 0.0355 - 5 | 0.0354 - 5 |
| convnext_small | 0.8360 | 0.0583 | 0.0479 | 0.0402 | 0.0402 | 0.0379 - 3 | 0.0376 - 5 |
| convnext_tiny | 0.8249 | 0.0617 | 0.0511 | 0.0454 | 0.0450 | 0.0417 - 5 | 0.0415 - 6 |
| densenet121 | 0.7439 | 0.0782 | 0.0784 | 0.0781 | 0.0821 | 0.0781 - 0 | 0.0797 - 0 |
| densenet161 | 0.7715 | 0.0665 | 0.0672 | 0.0665 | 0.0691 | 0.0650 - 3 | 0.0658 - 7 |
| densenet169 | 0.7555 | 0.0728 | 0.0734 | 0.0728 | 0.0760 | 0.0721 - 1 | 0.0730 - 7 |
| densenet201 | 0.7683 | 0.0675 | 0.0679 | 0.0674 | 0.0702 | 0.0665 - 3 | 0.0675 - 7 |
| efficientnet_b0 | 0.7768 | 0.0684 | 0.0655 | 0.0656 | 0.0663 | 0.0615 - 4 | 0.0616 - 6 |
| efficientnet_b1 | 0.7977 | 0.0616 | 0.0560 | 0.0548 | 0.0561 | 0.0534 - 4 | 0.0532 - 6 |
| efficientnet_b2 | 0.8054 | 0.0610 | 0.0562 | 0.0529 | 0.0539 | 0.0501 - 5 | 0.0500 - 6 |
| efficientnet_b3 | 0.8193 | 0.0587 | 0.0512 | 0.0470 | 0.0467 | 0.0443 - 5 | 0.0441 - 6 |
| efficientnet_b4 | 0.8333 | 0.0604 | 0.0474 | 0.0413 | 0.0415 | 0.0404 - 3 | 0.0399 - 7 |
| efficientnet_b5 | 0.8334 | 0.0504 | 0.0447 | 0.0406 | 0.0411 | 0.0386 - 4 | 0.0384 - 6 |
| efficientnet_b6 | 0.8389 | 0.0477 | 0.0422 | 0.0393 | 0.0389 | 0.0393 - 0 | 0.0367 - 0 |
| efficientnet_b7 | 0.8406 | 0.0500 | 0.0431 | 0.0374 | 0.0377 | 0.0360 - 3 | 0.0358 - 6 |
| efficientnet_v2_l | 0.8574 | 0.0431 | 0.0379 | 0.0335 | 0.0336 | 0.0317 - 5 | 0.0316 - 6 |
| efficientnet_v2_m | 0.8504 | 0.0513 | 0.0424 | 0.0362 | 0.0360 | 0.0345 - 3 | 0.0339 - 5 |
| efficientnet_v2_s | 0.8413 | 0.0465 | 0.0412 | 0.0376 | 0.0380 | 0.0360 - 4 | 0.0359 - 5 |
| googlenet | 0.6970 | 0.1056 | 0.1034 | 0.1030 | 0.1088 | 0.1023 - 3 | 0.1027 - 6 |
| inception_v3 | 0.7722 | 0.0736 | 0.0713 | 0.0676 | 0.0686 | 0.0641 - 4 | 0.0639 - 5 |
| maxvit_t | 0.8362 | 0.0476 | 0.0445 | 0.0402 | 0.0402 | 0.0383 - 5 | 0.0381 - 5 |
| mnasnet0_5 | 0.6769 | 0.1178 | 0.1146 | 0.1141 | 0.1204 | 0.1121 - 3 | 0.1121 - 7 |
| mnasnet0_75 | 0.7110 | 0.1207 | 0.0980 | 0.0970 | 0.1011 | 0.0939 - 3 | 0.0941 - 6 |
| mnasnet1_0 | 0.7349 | 0.0802 | 0.0807 | 0.0803 | 0.0843 | 0.0802 - 0 | 0.0821 - 0 |
| mnasnet1_3 | 0.7640 | 0.0975 | 0.0734 | 0.0706 | 0.0723 | 0.0676 - 4 | 0.0676 - 6 |
| mobilenet_v2 | 0.7211 | 0.1090 | 0.0914 | 0.0908 | 0.0951 | 0.0889 - 4 | 0.0891 - 6 |
| mobilenet_v3_large | 0.7529 | 0.0801 | 0.0752 | 0.0746 | 0.0783 | 0.0734 - 5 | 0.0734 - 6 |
| mobilenet_v3_small | 0.6770 | 0.1099 | 0.1103 | 0.1098 | 0.1162 | 0.1097 - 0 | 0.1129 - 0 |

| | | | | | | | |
|---|---|---|---|---|---|---|---|
| regnet_x_16gf | 0.8266 | 0.0517 | 0.0476 | 0.0434 | 0.0435 | 0.0414 - 4 | 0.0413 - 5 |
| regnet_x_1_6gf | 0.7962 | 0.0818 | 0.0605 | 0.0568 | 0.0577 | 0.0539 - 4 | 0.0538 - 6 |
| regnet_x_32gf | 0.8294 | 0.0523 | 0.0485 | 0.0421 | 0.0425 | 0.0406 - 3 | 0.0400 - 5 |
| regnet_x_3_2gf | 0.8111 | 0.0646 | 0.0558 | 0.0514 | 0.0518 | 0.0481 - 5 | 0.0482 - 6 |
| regnet_x_400mf | 0.7483 | 0.1009 | 0.0797 | 0.0779 | 0.0805 | 0.0743 - 4 | 0.0745 - 6 |
| regnet_x_800mf | 0.7748 | 0.0929 | 0.0696 | 0.0668 | 0.0684 | 0.0626 - 4 | 0.0626 - 5 |
| regnet_x_8gf | 0.8166 | 0.0563 | 0.0513 | 0.0470 | 0.0474 | 0.0452 - 3 | 0.0448 - 5 |
| regnet_y_128gf | 0.8819 | 0.0244 | 0.0245 | 0.0238 | 0.0242 | 0.0234 - 4 | 0.0233 - 7 |
| regnet_y_16gf | 0.8284 | 0.0591 | 0.0511 | 0.0426 | 0.0422 | 0.0398 - 4 | 0.0398 - 5 |
| regnet_y_1_6gf | 0.8085 | 0.0646 | 0.0559 | 0.0522 | 0.0524 | 0.0485 - 4 | 0.0485 - 5 |
| regnet_y_32gf | 0.8332 | 0.0517 | 0.0458 | 0.0405 | 0.0406 | 0.0384 - 3 | 0.0382 - 5 |
| regnet_y_3_2gf | 0.8189 | 0.0554 | 0.0499 | 0.0471 | 0.0476 | 0.0447 - 6 | 0.0448 - 5 |
| regnet_y_400mf | 0.7578 | 0.0861 | 0.0771 | 0.0751 | 0.0775 | 0.0707 - 4 | 0.0708 - 5 |
| regnet_y_800mf | 0.7881 | 0.0707 | 0.0624 | 0.0601 | 0.0617 | 0.0566 - 4 | 0.0565 - 5 |
| regnet_y_8gf | 0.8273 | 0.0518 | 0.0461 | 0.0426 | 0.0427 | 0.0405 - 5 | 0.0404 - 5 |
| resnet101 | 0.8185 | 0.0607 | 0.0533 | 0.0475 | 0.0476 | 0.0450 - 5 | 0.0447 - 5 |
| resnet152 | 0.8225 | 0.0576 | 0.0502 | 0.0441 | 0.0444 | 0.0422 - 4 | 0.0420 - 5 |
| resnet18 | 0.6971 | 0.1017 | 0.1021 | 0.1015 | 0.1069 | 0.1016 - 1 | 0.1039 - 7 |
| resnet34 | 0.7326 | 0.0830 | 0.0834 | 0.0831 | 0.0875 | 0.0829 - 2 | 0.0840 - 7 |
| resnet50 | 0.8082 | 0.0751 | 0.0560 | 0.0509 | 0.0514 | 0.0486 - 4 | 0.0484 - 5 |
| resnext101_32x8d | 0.8276 | 0.0820 | 0.0556 | 0.0450 | 0.0446 | 0.0413 - 4 | 0.0413 - 6 |
| resnext101_64x4d | 0.8316 | 0.0754 | 0.0508 | 0.0421 | 0.0418 | 0.0397 - 3 | 0.0392 - 6 |
| resnext50_32x4d | 0.8116 | 0.0645 | 0.0563 | 0.0502 | 0.0507 | 0.0478 - 5 | 0.0474 - 5 |
| shufflenet_v2_x0_5 | 0.6052 | 0.1575 | 0.1583 | 0.1571 | 0.1642 | 0.1561 - 4 | 0.1565 - 7 |
| shufflenet_v2_x1_0 | 0.6924 | 0.1031 | 0.1041 | 0.1030 | 0.1067 | 0.1019 - 4 | 0.1019 - 7 |
| shufflenet_v2_x1_5 | 0.7299 | 0.1061 | 0.0890 | 0.0879 | 0.0916 | 0.0854 - 4 | 0.0855 - 6 |
| shufflenet_v2_x2_0 | 0.7616 | 0.0895 | 0.0733 | 0.0712 | 0.0730 | 0.0680 - 4 | 0.0679 - 6 |
| squeezenet1_0 | 0.5804 | 0.1777 | 0.1784 | 0.1770 | 0.1865 | 0.1769 - 0 | 0.1906 - 0 |
| squeezenet1_1 | 0.5817 | 0.1735 | 0.1740 | 0.1731 | 0.1834 | 0.1731 - 0 | 0.1861 - 0 |
| swin_b | 0.8353 | 0.0561 | 0.0508 | 0.0409 | 0.0410 | 0.0391 - 3 | 0.0384 - 5 |
| swin_s | 0.8316 | 0.0507 | 0.0479 | 0.0428 | 0.0428 | 0.0405 - 4 | 0.0402 - 5 |
| swin_t | 0.8143 | 0.0547 | 0.0511 | 0.0486 | 0.0494 | 0.0465 - 4 | 0.0463 - 5 |
| swin_v2_b | 0.8412 | 0.0496 | 0.0456 | 0.0389 | 0.0392 | 0.0371 - 4 | 0.0369 - 5 |
| swin_v2_s | 0.8365 | 0.0488 | 0.0447 | 0.0394 | 0.0396 | 0.0377 - 4 | 0.0375 - 5 |
| swin_v2_t | 0.8204 | 0.0526 | 0.0485 | 0.0456 | 0.0462 | 0.0437 - 4 | 0.0436 - 5 |
| vgg11 | 0.6909 | 0.1030 | 0.1032 | 0.1029 | 0.1089 | 0.1028 - 0 | 0.1085 - 0 |

| | | | | | | |
|---|---|---|---|---|---|---|
| vgg11_bn | 0.7034 | 0.0961 | 0.0964 | 0.0960 | 0.1015 | 0.0960 - 0 | 0.1019 - 0 |
| vgg13 | 0.6988 | 0.0983 | 0.0985 | 0.0982 | 0.1036 | 0.0982 - 0 | 0.1033 - 0 |
| vgg13_bn | 0.7151 | 0.0902 | 0.0905 | 0.0901 | 0.0954 | 0.0901 - 0 | 0.0948 - 0 |
| vgg16 | 0.7156 | 0.0890 | 0.0892 | 0.0889 | 0.0940 | 0.0889 - 0 | 0.0931 - 0 |
| vgg16_bn | 0.7335 | 0.0804 | 0.0808 | 0.0804 | 0.0845 | 0.0803 - 0 | 0.0837 - 0 |
| vgg19 | 0.7232 | 0.0853 | 0.0856 | 0.0853 | 0.0903 | 0.0853 - 0 | 0.0890 - 0 |
| vgg19_bn | 0.7420 | 0.0772 | 0.0775 | 0.0772 | 0.0810 | 0.0771 - 0 | 0.0808 - 0 |
| vit_b_16 | 0.8102 | 0.0595 | 0.0553 | 0.0502 | 0.0505 | 0.0477 - 4 | 0.0475 - 5 |
| vit_b_32 | 0.7592 | 0.0792 | 0.0754 | 0.0714 | 0.0723 | 0.0675 - 4 | 0.0674 - 5 |
| vit_h_14 | 0.8848 | 0.0253 | 0.0248 | 0.0235 | 0.0238 | 0.0229 - 4 | 0.0229 - 6 |
| vit_l_16 | 0.7969 | 0.0628 | 0.0610 | 0.0555 | 0.0560 | 0.0520 - 4 | 0.0521 - 4 |
| vit_l_32 | 0.7696 | 0.0784 | 0.0750 | 0.0678 | 0.0678 | 0.0624 - 4 | 0.0626 - 5 |
| wide_resnet101_2 | 0.8246 | 0.0603 | 0.0522 | 0.0450 | 0.0445 | 0.0419 - 5 | 0.0417 - 5 |
| wide_resnet50_2 | 0.8155 | 0.0778 | 0.0562 | 0.0486 | 0.0490 | 0.0458 - 4 | 0.0456 - 6 |
| efficientnetv2_xl | 0.8555 | 0.0698 | 0.0578 | 0.0384 | 0.0368 | 0.0338 - 4 | 0.0334 - 6 |
| vit_l_16_384 | 0.8702 | 0.0265 | 0.0265 | 0.0264 | 0.0273 | 0.0263 - 0 | 0.0269 - 0 |
| vit_b_16_sam | 0.8016 | 0.0487 | 0.0487 | 0.0485 | 0.0497 | 0.0485 - 0 | 0.0487 - 0 |
| vit_b_32_sam | 0.7368 | 0.0761 | 0.0760 | 0.0759 | 0.0784 | 0.0758 - 0 | 0.0762 - 0 |

Table 10: AUROC for all models evaluated on ImageNet

| Model | Accuracy[%] | Method | | | | | |
|---|---|---|---|---|---|---|---|
| | | MSP | MSP-TS-NLL | MSP-TS-AURC | LogitsMargin | MSP-pNorm - $p^*$ | MaxLogit-pNorm - $p^*$ |
| alexnet | 0.5657 | 0.8483 | 0.8475 | 0.8483 | 0.8181 | 0.8482 - 0 | 0.8391 - 0 |
| convnext_base | 0.8402 | 0.8255 | 0.8537 | 0.8672 | 0.8652 | 0.8769 - 4 | 0.8771 - 5 |
| convnext_large | 0.8438 | 0.8257 | 0.8496 | 0.8700 | 0.8686 | 0.8793 - 5 | 0.8796 - 5 |
| convnext_small | 0.8360 | 0.8271 | 0.8560 | 0.8702 | 0.8678 | 0.8780 - 3 | 0.8801 - 5 |
| convnext_tiny | 0.8249 | 0.8247 | 0.8560 | 0.8659 | 0.8633 | 0.8781 - 5 | 0.8786 - 6 |
| densenet121 | 0.7439 | 0.8604 | 0.8596 | 0.8604 | 0.8454 | 0.8603 - 0 | 0.8564 - 0 |
| densenet161 | 0.7715 | 0.8635 | 0.8612 | 0.8633 | 0.8520 | 0.8668 - 3 | 0.8643 - 7 |
| densenet169 | 0.7555 | 0.8654 | 0.8634 | 0.8654 | 0.8520 | 0.8663 - 1 | 0.8631 - 7 |
| densenet201 | 0.7683 | 0.8628 | 0.8616 | 0.8630 | 0.8512 | 0.8647 - 3 | 0.8619 - 7 |
| efficientnet_b0 | 0.7768 | 0.8570 | 0.8642 | 0.8581 | 0.8535 | 0.8704 - 4 | 0.8701 - 6 |
| efficientnet_b1 | 0.7977 | 0.8538 | 0.8672 | 0.8676 | 0.8584 | 0.8711 - 4 | 0.8713 - 6 |
| efficientnet_b2 | 0.8054 | 0.8509 | 0.8628 | 0.8632 | 0.8592 | 0.8740 - 5 | 0.8740 - 6 |
| efficientnet_b3 | 0.8193 | 0.8435 | 0.8636 | 0.8701 | 0.8660 | 0.8770 - 5 | 0.8775 - 6 |
| efficientnet_b4 | 0.8333 | 0.8213 | 0.8600 | 0.8697 | 0.8668 | 0.8739 - 3 | 0.8750 - 7 |
| efficientnet_b5 | 0.8334 | 0.8549 | 0.8691 | 0.8724 | 0.8693 | 0.8809 - 4 | 0.8817 - 6 |
| efficientnet_b6 | 0.8389 | 0.8575 | 0.8714 | 0.8750 | 0.8704 | 0.8750 - 0 | 0.8811 - 0 |
| efficientnet_b7 | 0.8406 | 0.8509 | 0.8676 | 0.8761 | 0.8730 | 0.8814 - 3 | 0.8827 - 6 |
| efficientnet_v2_l | 0.8574 | 0.8467 | 0.8645 | 0.8724 | 0.8700 | 0.8796 - 5 | 0.8798 - 6 |
| efficientnet_v2_m | 0.8504 | 0.8251 | 0.8549 | 0.8696 | 0.8680 | 0.8750 - 3 | 0.8779 - 5 |
| efficientnet_v2_s | 0.8413 | 0.8591 | 0.8718 | 0.8740 | 0.8707 | 0.8810 - 4 | 0.8815 - 5 |
| googlenet | 0.6970 | 0.8490 | 0.8542 | 0.8550 | 0.8361 | 0.8563 - 3 | 0.8552 - 6 |
| inception_v3 | 0.7722 | 0.8491 | 0.8548 | 0.8605 | 0.8537 | 0.8700 - 4 | 0.8702 - 5 |
| maxvit_t | 0.8362 | 0.8594 | 0.8658 | 0.8698 | 0.8672 | 0.8776 - 5 | 0.8781 - 5 |
| mnasnet0_5 | 0.6769 | 0.8465 | 0.8534 | 0.8536 | 0.8333 | 0.8577 - 3 | 0.8574 - 7 |
| mnasnet0_75 | 0.7110 | 0.8025 | 0.8510 | 0.8526 | 0.8377 | 0.8592 - 3 | 0.8584 - 6 |
| mnasnet1_0 | 0.7349 | 0.8659 | 0.8643 | 0.8658 | 0.8506 | 0.8658 - 0 | 0.8601 - 0 |
| mnasnet1_3 | 0.7640 | 0.7942 | 0.8554 | 0.8592 | 0.8501 | 0.8665 - 4 | 0.8662 - 6 |
| mobilenet_v2 | 0.7211 | 0.8167 | 0.8554 | 0.8559 | 0.8399 | 0.8600 - 4 | 0.8593 - 6 |
| mobilenet_v3_large | 0.7529 | 0.8519 | 0.8625 | 0.8620 | 0.8464 | 0.8638 - 5 | 0.8636 - 6 |
| mobilenet_v3_small | 0.6770 | 0.8621 | 0.8611 | 0.8622 | 0.8415 | 0.8622 - 0 | 0.8547 - 0 |

| | | | | | | |
|---|---|---|---|---|---|---|
| regnet_x_16gf | 0.8266 | 0.8553 | 0.8664 | 0.8703 | 0.8675 | 0.8777 - 4 | 0.8781 - 5 |
| regnet_x_1_6gf | 0.7962 | 0.7991 | 0.8584 | 0.8627 | 0.8575 | 0.8718 - 4 | 0.8720 - 6 |
| regnet_x_32gf | 0.8294 | 0.8551 | 0.8643 | 0.8717 | 0.8685 | 0.8766 - 3 | 0.8794 - 5 |
| regnet_x_3_2gf | 0.8111 | 0.8340 | 0.8583 | 0.8634 | 0.8592 | 0.8722 - 5 | 0.8718 - 6 |
| regnet_x_400mf | 0.7483 | 0.8133 | 0.8583 | 0.8583 | 0.8457 | 0.8660 - 4 | 0.8654 - 6 |
| regnet_x_800mf | 0.7748 | 0.7981 | 0.8562 | 0.8593 | 0.8503 | 0.8691 - 4 | 0.8688 - 5 |
| regnet_x_8gf | 0.8166 | 0.8532 | 0.8656 | 0.8698 | 0.8660 | 0.8750 - 3 | 0.8774 - 5 |
| regnet_y_128gf | 0.8819 | 0.8836 | 0.8832 | 0.8835 | 0.8804 | 0.8857 - 4 | 0.8862 - 7 |
| regnet_y_16gf | 0.8284 | 0.8397 | 0.8590 | 0.8715 | 0.8697 | 0.8809 - 4 | 0.8810 - 5 |
| regnet_y_1_6gf | 0.8085 | 0.8396 | 0.8616 | 0.8660 | 0.8588 | 0.8742 - 4 | 0.8742 - 5 |
| regnet_y_32gf | 0.8332 | 0.8488 | 0.8642 | 0.8714 | 0.8684 | 0.8801 - 3 | 0.8810 - 5 |
| regnet_y_3_2gf | 0.8189 | 0.8522 | 0.8654 | 0.8672 | 0.8615 | 0.8739 - 6 | 0.8739 - 5 |
| regnet_y_400mf | 0.7578 | 0.8386 | 0.8575 | 0.8574 | 0.8448 | 0.8667 - 4 | 0.8661 - 5 |
| regnet_y_800mf | 0.7881 | 0.8413 | 0.8610 | 0.8613 | 0.8521 | 0.8715 - 4 | 0.8711 - 5 |
| regnet_y_8gf | 0.8273 | 0.8526 | 0.8683 | 0.8718 | 0.8687 | 0.8794 - 5 | 0.8797 - 5 |
| resnet101 | 0.8185 | 0.8423 | 0.8604 | 0.8665 | 0.8631 | 0.8753 - 5 | 0.8758 - 5 |
| resnet152 | 0.8225 | 0.8465 | 0.8652 | 0.8722 | 0.8691 | 0.8799 - 4 | 0.8808 - 5 |
| resnet18 | 0.6971 | 0.8573 | 0.8561 | 0.8577 | 0.8398 | 0.8573 - 1 | 0.8521 - 7 |
| resnet34 | 0.7326 | 0.8619 | 0.8608 | 0.8618 | 0.8456 | 0.8621 - 2 | 0.8592 - 7 |
| resnet50 | 0.8082 | 0.8060 | 0.8602 | 0.8658 | 0.8617 | 0.8744 - 4 | 0.8745 - 5 |
| resnext101_32x8d | 0.8276 | 0.7680 | 0.8447 | 0.8654 | 0.8635 | 0.8768 - 4 | 0.8769 - 6 |
| resnext101_64x4d | 0.8316 | 0.7796 | 0.8534 | 0.8701 | 0.8684 | 0.8777 - 3 | 0.8799 - 6 |
| resnext50_32x4d | 0.8116 | 0.8360 | 0.8569 | 0.8644 | 0.8606 | 0.8735 - 5 | 0.8743 - 5 |
| shufflenet_v2_x0_5 | 0.6052 | 0.8513 | 0.8498 | 0.8519 | 0.8319 | 0.8534 - 4 | 0.8524 - 7 |
| shufflenet_v2_x1_0 | 0.6924 | 0.8602 | 0.8578 | 0.8602 | 0.8469 | 0.8629 - 4 | 0.8628 - 7 |
| shufflenet_v2_x1_5 | 0.7299 | 0.8123 | 0.8518 | 0.8536 | 0.8390 | 0.8594 - 4 | 0.8587 - 6 |
| shufflenet_v2_x2_0 | 0.7616 | 0.8170 | 0.8588 | 0.8617 | 0.8524 | 0.8693 - 4 | 0.8693 - 6 |
| squeezenet1_0 | 0.5804 | 0.8424 | 0.8410 | 0.8437 | 0.8182 | 0.8436 - 0 | 0.8177 - 0 |
| squeezenet1_1 | 0.5817 | 0.8486 | 0.8476 | 0.8492 | 0.8227 | 0.8492 - 0 | 0.8254 - 0 |
| swin_b | 0.8353 | 0.8432 | 0.8548 | 0.8678 | 0.8653 | 0.8757 - 3 | 0.8788 - 5 |
| swin_s | 0.8316 | 0.8554 | 0.8615 | 0.8667 | 0.8631 | 0.8752 - 4 | 0.8758 - 5 |
| swin_t | 0.8143 | 0.8588 | 0.8668 | 0.8683 | 0.8612 | 0.8750 - 4 | 0.8751 - 5 |
| swin_v2_b | 0.8412 | 0.8516 | 0.8604 | 0.8672 | 0.8645 | 0.8762 - 4 | 0.8766 - 5 |
| swin_v2_s | 0.8365 | 0.8592 | 0.8680 | 0.8733 | 0.8703 | 0.8805 - 4 | 0.8813 - 5 |
| swin_v2_t | 0.8204 | 0.8593 | 0.8683 | 0.8696 | 0.8647 | 0.8769 - 4 | 0.8772 - 5 |
| vgg11 | 0.6909 | 0.8609 | 0.8602 | 0.8612 | 0.8410 | 0.8611 - 0 | 0.8480 - 0 |

| | | | | | | |
|---|---|---|---|---|---|---|
| vgg11_bn | 0.7034 | 0.8626 | 0.8617 | 0.8627 | 0.8444 | 0.8626 - 0 | 0.8491 - 0 |
| vgg13 | 0.6988 | 0.8619 | 0.8613 | 0.8622 | 0.8442 | 0.8621 - 0 | 0.8502 - 0 |
| vgg13_bn | 0.7151 | 0.8634 | 0.8623 | 0.8635 | 0.8451 | 0.8634 - 0 | 0.8522 - 0 |
| vgg16 | 0.7156 | 0.8661 | 0.8652 | 0.8662 | 0.8478 | 0.8661 - 0 | 0.8556 - 0 |
| vgg16_bn | 0.7335 | 0.8679 | 0.8665 | 0.8680 | 0.8525 | 0.8679 - 0 | 0.8591 - 0 |
| vgg19 | 0.7232 | 0.8656 | 0.8647 | 0.8656 | 0.8479 | 0.8656 - 0 | 0.8565 - 0 |
| vgg19_bn | 0.7420 | 0.8654 | 0.8641 | 0.8654 | 0.8510 | 0.8654 - 0 | 0.8560 - 0 |
| vit_b_16 | 0.8102 | 0.8559 | 0.8637 | 0.8680 | 0.8621 | 0.8756 - 4 | 0.8762 - 5 |
| vit_b_32 | 0.7592 | 0.8559 | 0.8631 | 0.8623 | 0.8556 | 0.8742 - 4 | 0.8743 - 5 |
| vit_h_14 | 0.8848 | 0.8754 | 0.8777 | 0.8816 | 0.8791 | 0.8848 - 4 | 0.8850 - 6 |
| vit_l_16 | 0.7969 | 0.8587 | 0.8618 | 0.8642 | 0.8603 | 0.8767 - 4 | 0.8767 - 4 |
| vit_l_32 | 0.7696 | 0.8542 | 0.8591 | 0.8631 | 0.8561 | 0.8760 - 4 | 0.8750 - 5 |
| wide_resnet101_2 | 0.8246 | 0.8384 | 0.8592 | 0.8691 | 0.8665 | 0.8787 - 5 | 0.8790 - 5 |
| wide_resnet50_2 | 0.8155 | 0.7920 | 0.8524 | 0.8646 | 0.8611 | 0.8749 - 4 | 0.8752 - 6 |
| efficientnetv2_xl | 0.8555 | 0.7732 | 0.8109 | 0.8587 | 0.8604 | 0.8708 - 4 | 0.8740 - 6 |
| vit_l_16_384 | 0.8702 | 0.8857 | 0.8855 | 0.8862 | 0.8801 | 0.8861 - 0 | 0.8837 - 0 |
| vit_b_16_sam | 0.8016 | 0.8825 | 0.8827 | 0.8833 | 0.8770 | 0.8832 - 0 | 0.8825 - 0 |
| vit_b_32_sam | 0.7368 | 0.8754 | 0.8758 | 0.8762 | 0.8661 | 0.8761 - 0 | 0.8755 - 0 |

