# OpenReview forum: "How to fix a broken confidence estimator: Evaluating post-hoc methods for selective classification with deep neural networks"
_ICLR.cc/2024/Conference — Submitted to ICLR 2024_

### Official Review · Reviewer_TapJ · 2023-10-27

**Soundness:** 3 good
**Presentation:** 3 good
**Contribution:** 3 good
**Rating:** 6
**Confidence:** 4

**Summary:**

This paper addresses the problem of how to improve selective classification (i.e. predicting only on some subset of the data so as to minimize mistakes) via post-hoc modifications to the confidence predictions of trained models (i.e. modifying the logits/softmax outputs). The main contributions of this work are a detailed comparison of different post-hoc methods applied to various model confidence estimation procedures, along with a new metric for performing this comparison (normalized AURC).

**Strengths:**

- **Originality:** Although I am not fully familiar with the line of work on selective classification, the idea of NAURC is a nice (and to me, new) extension of AURC that allows comparison across tasks as the authors mention. On the post-training calibration side, the main new idea is logit $p$-normalization with tunable $p$ and scaling, as a sort of simple version of input-dependent temperature scaling, and this also seems new to me.
- **Quality:** The quality of the work is good; the problem is well-motivated, the considered confidence estimators for models are comprehensive, and the evaluation is done well.
- **Clarity:** Overall the paper is easy to follow and organized well.
- **Significance:** While the actual novelty of the $p$-norm confidence tuning seems low given prior work, I believe the empirical results - particularly the data efficiency of this approach - seems to be non-trivially useful/significant. Additionally, the empirical comparison done in this work will certainly be useful for those interested in applying post-training methods to selective classification.

**Weaknesses:**

## Main Weaknesses
1. **Insufficient comparison to post-training methods.** My main concern with the evaluation in this work is that the only two confidence tuning approaches considered are standard temperature scaling (TS) and $p$-norm tuning, and the conclusion is that $p$-norm tuning can be better for the tasks considered. However, there are several similarly simple (in terms of parameters) approaches introduced in [1], which the authors themselves reference when proposing $p$-norm tuning, which are not compared to in the main results of the paper. As a result, it is difficult to determine how much of a benefit there really is from the proposed $p$-norm tuning.
2. **Discussion of related work on calibration.** While the authors do a good job of discussing prior work on selective classification, there is very little discussion of work on post-training calibration methods. Namely, I think the work would benefit from discussion of modifications to TS/related strategies with more justification for why comparing $p$-norm normalization with vanilla TS is sufficient for making their points.

## Minor Comments
- Typo on page 8 s/"Logits Marging"/"Logits Margin"
- Table 7 in the Appendix (with the comprehensive results) could be presented better.

## Recommendation
While I have some reservations about the novelty of the methods in the work, I think there are several useful ideas (NAURC) and insights in the paper. Particularly, I find the data-efficiency observations to be of note (as discussed in Appendix E and at the top of page 8 of the main paper), and would have liked to see more related discussion in the main paper - this seems to be a very useful attribute and good point of comparison. As a result, I lean accept for this paper and recommend **weak accept**.

**Questions:**

- In Section 3.2.2, the authors write: "a useful property of MSP-TS (but not MSP-TS-NLL) is that it can never have worse performance than the MSP baseline"; I do not see why this is true. As long as we do TS on held-out calibration/validation data, it is certainly still possible that any metric could be worse on test data, unless I'm misunderstanding what the authors mean by MSP-TS here.
- What is the best choice of $p$ found by tuning $p$-norm? It would be useful to include this in the main discussion, and whether there is actually some significant benefit to tuning $p$ instead of just using $p = 2$.
- In Table 3, it would be useful to include error bounds (like 1 std range) since you are reporting averages across all models here to better get a sense of the improvement.

---

> ### Author Response · Authors · 2023-11-19
> **Response to Reviewer TapJ**
>
> Thank you very much for your constructive feedback. Please find our responses below. We hope they are satisfactory and you will consider raising your score.
>
> >**W1:** Insufficient comparison to post-training methods. (...) there are several similarly simple (in terms of parameters) approaches introduced in [1], (...) which are not compared to in the main results of the paper.
>
> Thank you for your suggestion. We included a new Appendix H with a comparison to other existing tunable methods, including the main method proposed by Balanya et al. (2023) (we assume this is the [1] you are referring to). As can be seen from the table below, MaxLogit-pNorm showed superior performance to all of these methods, while requiring much less hyperparameter tuning.
>
> | Method | APG-NAURC |
> |-|-|
> | ETS  (Zhang et al., 2020) | 0.05776 |
> | BK (Boursinos & Koutsoukos, 2022) | 0.02504 |
> | HTS (Balanya et al., 2023) | 0.06286 |
> | MaxLogit-pNorm | **0.06837** |
>
> >**W2:** Discussion of related work on calibration. (...)  I think the work would benefit from discussion of modifications to TS/related strategies with more justification for why comparing $p$-norm normalization with vanilla TS is sufficient for making their points.
>
> Thank you for this suggestion. We have included in the new Appendix H a review of the methods mentioned in the above table as well as a discussion of post-hoc calibration methods with a large number of tunable parameters and a justification for not including them in our evaluation.
>
> >**Q1:** In Section 3.2.2, the authors write: "a useful property of MSP-TS (but not MSP-TS-NLL) is that it can never have worse performance than the MSP baseline"; I do not see why this is true. As long as we do TS on held-out calibration/validation data, it is certainly still possible that any metric could be worse on test data.
>
> The reviewer is correct; thanks for pointing this out. We have fixed the sentence as: "A useful property of MSP-TS (but not MSP-TS-NLL) is that, *in the infinite-sample setting,* it can never have a worse performance than the MSP baseline, as long as T = 1 is included in the search space."
>
> >**Q2:** What is the best choice of $p$ found by tuning $p$-norm? It would be useful to include this in the main discussion, and whether there is actually some significant benefit to tuning $p$ instead of just using $p$=2.
>
> For MaxLogit-pNorm, the best choice of $p$ for each model is shown in the legend of Fig. 3, where we can see it is typically between 4 and 6. It is also given in the appendix tables with the full results for MSP-pNorm and MaxLogit-pNorm. We have now included a new Appendix G with a comparison between tunable and fixed $p$ for these two methods and included a pointer to these results in Section 4.1. As can be seen in the table below, there is a significant benefit of tuning $p$ compared to just using $p=2$. Note that the statistical significance of these benefits is discussed in the response to the next question.
>
> |$p$|MaxLogit-pNorm|MSP-pNorm|
> |-|-|-|
> |0|0.00000|0.05776|
> |1|0.00197|0.06030|
> |2|0.01471|0.06501|
> |3|0.05039|0.06714|
> |4|0.06416|0.06798|
> |5|0.06778|0.06787|
> |6|0.06785|0.06741|
> |7|0.06654|0.06703|
> |Tunable|0.06837|0.06781|
>
> >**Q3:** In Table 3, it would be useful to include error bounds (like 1 std range) since you are reporting averages across all models here to better get a sense of the improvement.
>
> Please note that we are reporting averages across models simply as a way to summarize performance, but it would be potentially misleading to report error bars for these averages, as the choice of a model is not a random variable, but rather a specific experiment scenario. In our data efficiency experiments (Figures 10 and 11 in the new version), we had already shown 10-90 percentile regions, which are obtained by randomly varying the validation-test split, for a specific model ResNet50. We have now included a new Appendix I where we perform a similar analysis for all models. Figures 12 and 13 (shown below) show +-1 standard deviation around the mean, with models sorted by NAURC gain over MSP for MaxLogit-pNorm (similarly to Fig. 2). We can see that, when the validation set has 5000 samples, the gains of MaxLogit-pNorm are statistically significant compared to MSP-TS and MSP-pNorm with fixed $p=2$, but not compared to MSP-pNorm with tunable $p$, which was already suggested by Fig. 2. However, when the validation set is reduced to 1000 samples, MaxLogit-pNorm outperforms all three methods, which was already suggested by the data efficiency experiments. Moreover, we can see that MaxLogit-pNorm is the most stable among these methods in terms of variance of gains.
>
> [Click here to see comparison plots with error bounds](https://postimg.cc/5HZTFxgm)
>
> We hope we have addressed all of the reviewer’s concerns satisfactorily and would be happy to answer any other questions.

---

> ### Author Response · Authors · 2023-11-21
> **Gentle reminder**
>
> Thank you again for your constructive feedback. We believe we have addressed all your concerns.
>
> We would like to remind you that discussion phase is ending soon. We hope you will consider raising your score. If there are still any reasons you think this work does not deserve a higher score, please let us know and we will be happy to address them.

---

> > ### Comment · Reviewer_TapJ · 2023-11-21
> >
> > Thank you for the very detailed response as well as the summary of revisions. I have no further questions and believe my original questions to have been appropriately addressed. I now am more in favor of acceptance (I would consider my score between a 6 and an 8) and am happy to defend the paper in discussions. My only reservation in increasing the score of the paper still stems from novelty; I acknowledge the authors' claims regarding the novelty of their contributions, but the overall method still does not feel sufficiently new relative to prior work (although the experimental results are indeed impressive).

---

> > > ### Author Response · Authors · 2023-11-22
> > > **Thanks**
> > >
> > > Thanks so much for your feedback and support of our paper!

---

### Official Review · Reviewer_3CxY · 2023-10-29

**Soundness:** 3 good
**Presentation:** 2 fair
**Contribution:** 2 fair
**Rating:** 5
**Confidence:** 4

**Summary:**

This paper considers the problem of learning selective classification. In particular, it explores possibilities in designing better confidence estimators. In experiments, this paper considers multiple known confidence estimators along with multiple logit transformations. Also, for unified comparison, this paper proposes a new metric, called normalized area under risk coverage (NAURC). Based on this new metric, a simple p-norm normalization of the logits along with the standard maximum logit  as a confidence estimator leads to considerable gain in selective classification performance.

**Strengths:**

This paper proposes a new evaluation metric, i.e., NAURC, and provides a new finding (i.e., a simple p-norm normalization of the logits along with the standard maximum logit  as a confidence estimator is good for selective classification) from extensive empirical evaluation.

**Weaknesses:**

This paper explores an interesting issue on recovering “broken” confidence estimators, which is tightly related to a popular calibration problem, though I found that the conclusion of this paper is unclear.

1. The main corner is that the connection of the conclusion of this paper to the guaranteed risk control (i.e, (2) in Geifman & El-Yaniv (2017)) is unclear. NAURC can be a good intermediate metric, but this confidence estimator should be used along with a selective classifier. Then, the confidence estimator with a good NAURC should demonstrate its benefits along with a selective classifier (in terms of achieving a desired risk level) for in-distribution and out-of-distribution experiments.
2. I’m not quite motivated from Figure 1. What’s the reason that the left figure is bad? In terms of smoothness, yes it looks bad, but in constructing a guaranteed selective classifier, can it be the problem? Probably, adding some context would be useful in motivating readers.

**Questions:**

1. What’s the benefits of conclusion of this paper in terms of the performance of the final guaranteed risk controlled selective classifier by  Geifman & El-Yaniv (2017)?
2. What’s the reason that the left figure is bad in terms of constructing a guaranteed selective classifier?

---

> ### Author Response · Authors · 2023-11-15
> **Response to Reviewer 3CxY**
>
> Thank you very much for your constructive feedback. Please find our responses below.
>
> > I’m not quite motivated from Figure 1. What’s the reason that the left figure is bad in terms of constructing a guaranteed selective classifier? Probably, adding some context would be useful in motivating readers.
>
> We agree that it would be beneficial to add more context to Fig 1 and we thank the reviewer for pointing this out. We will replace the caption of Fig 1 with:
>
> *Figure 1: A comparison of RC curves made by three models selected in (Galil et al., 2023), including examples of highest (ViT-L/16-384) and lowest (EfficientNet-V2-XL) AUROC. An RC curve shows the tradeoff between risk (in this case, error rate) and coverage. The initial risk for any classifier is found at the 100% coverage point, where all predictions are accepted. Normally, the risk can be reduced by reducing coverage (which is done by increasing the selection threshold); for instance, a 2% error rate can be obtained at 36.2% coverage for the ViT-B/32-224-SAM model and at 61.9% coverage for the ViT-L/16-38 model. However, for the EfficientNet-V2-XL model, this error rate is not achievable at any coverage, since its RC curve is lower bounded by 5% risk. Moreover, this RC curve is actually non-monotonic, with an increasing risk as coverage is reduced, for low coverage. Fortunately, this apparent pathology in EfficientNet-V2-XL completely disappears after a simple post-hoc optimization of its confidence estimator, resulting in significantly improved selective classification performance. In particular, a 2% error rate can then be achieved at 55.3% coverage.*
>
> We hope this explanation clarifies why the red curve on the left figure is not desirable in terms of constructing a selective classifier with a low risk.
>
> > What’s the benefits of conclusion of this paper in terms of the performance of the final guaranteed risk controlled selective classifier by Geifman & El-Yaniv (2017)? The confidence estimator with a good NAURC should demonstrate its benefits along with a selective classifier (in terms of achieving a desired risk level) for in-distribution and out-of-distribution experiments.
>
> We agree that an important (and tangible) way to evaluate the performance of a selective classifier is to compute the maximum coverage allowed such that the corresponding selective risk is below some target level (this is essentially the definition of the SAC metric). But note that this information, for all risk levels, can be directly obtained from the RC curve: we choose a desired risk level on the y axis and then look for the rightmost point that crosses this level. Thus, the RC curve provides the evaluation suggested by the reviewer.
>
> In this context, some of the conclusions of our paper are:
> 1. Before post-hoc optimization, many models show pathological RC curves, where at some point the risk starts to increase as coverage decreases. But, after optimization, all RC curves start to look very similar to Fig 1, with the risk going to zero as coverage decreases. Further examples of RC curves can be found in Fig 5 for ImageNet (and Figs 8 and 9 for other datasets), but rather than plotting 84 similar-looking RC curves, we show in Fig 3(b) that, after post-hoc optimization, the AURC becomes extremely (Spearman-)correlated to the classifier accuracy, suggesting that better selective performance (for any risk level) can be obtained simply by choosing a model with better accuracy.
> 2. In order to give a more concrete evaluation in a distribution shift scenario, we show in Table 3 the maximum coverage allowed such that the accuracy is the same as the original accuracy of the model without distribution shift. We can see that, while coverage always drops with distribution shift (the RC curve moves up), it drops much less after post-hoc optimization is applied. This is not obvious a priori, since the confidence estimator is optimized with in-distribution data, so optimization could potentially be harmful to out-of-distribution generalization, but it is actually beneficial.
>
> Motivated by the reviewer's comment, in order to make point 1 above clearer, we will include Figure 7 in the Appendix D.5 showing the SAC metric for a target 98% accuracy for all 84 models on ImageNet, before and after post-hoc optimization. This figure shows that most models achieve significant coverage gains with post-hoc optimization. Moreover, for several models, achieving this target accuracy is virtually impossible without post-hoc optimization (except with coverage very close to 0%). Thus, post-hoc optimization is highly beneficial in achieving a desired target risk.
>
> For convenience, a few selected points are shown below (SAC = coverage for 98% accuracy):
>
> |Model|Accuracy[%]|SAC-Baseline[%]|SAC-Optimized[%]|
> |-|-|-|-|
> |VGG16|71.56|29.79|29.79|
> |ResNet50|80.82|0.17|45.58|
> |WideResNet50-2|81.55|0.14|47.78|
> |ConvNext-Base|84.02|0.22|53.14|
> |EfficientNetV2-XL|85.55|0.03|55.30|
> |Vit-h-14|88.48|61.34|65.26|

---

> > ### Comment · Reviewer_3CxY · 2023-12-01
> >
> > Thanks for the detailed response (and sorry for the late reply).
> >
> > On Figure 1: given the detailed context, I have better understanding on this plot. But, I'm still confused on why the "optimized" plot should be the "right/true" RC plot. I can see the the "optimized" plot is neat and nice, but the "baseline" is okay to me (as it is what it is). There is no definition on the "true" risk at a given coverage rate, and I believe that the added justification on the "right/true" RC plot is greatly beneficial.
> >
> > For this reason, I'll maintain my score (but, as I couldn't involve the discussion early, I won't fight for rejection).

---

> ### Author Response · Authors · 2023-11-21
> **Gentle reminder**
>
> Thank you again for your constructive feedback. We believe we have addressed all your concerns.
>
> We would like to remind you that discussion phase is ending soon. We hope you will consider raising your score. If there are still any reasons you think this work should not be accepted, please let us know and we will be happy to address them.

---

### Official Review · Reviewer_aDAg · 2023-10-29

**Soundness:** 2 fair
**Presentation:** 2 fair
**Contribution:** 2 fair
**Rating:** 6
**Confidence:** 4

**Summary:**

In this study, the authors present a confidence estimator for selective classification, aiming to optimize prediction accuracy by effectively filtering out uncertain samples (i.e. more likely to be misclassified). The proposed method includes 1) temperature scaling with AURC as the optimization objective, 2) logit normalization with learnable p (p from p-norm) and temperature t, 3) a normalized AURC as the evaluation metric which is monotonically related to AURC but avoids AURC's drawback of arbitrary value, 4) MSP fallback: check whether the estimator provides an improvement to the MSP baseline and, if not, then use the MSP instead. Compared to existing logit-adjusting strategies, the authors identify MaxLogit-pNorm as the clear winner in terms of NAURC.

**Strengths:**

Originality: The authors propose 1) a learnable version of logit normalization for selective classification; 2) a new metric normalized AURC to avoid the arbitrary value issue of AURC.

Clarity: In general, the paper is easy to understand.

Significance: 1) The method itself is simple and effective. 2) This study provides valuable insights into its application in selective classification, e.g., it shows that MaxLogit-pNorm has superior performance and also studies what kind of methods are more likely to benefit from logit normalization and what cannot.

**Weaknesses:**

Originality: The paper makes an incremental contribution to the field by extending the logit normalization to selective classification, initially introduced in "Mitigating Neural Network Overconfidence with Logit Normalization." The current study diverges from the previous work by two aspects: 1) The employed norm's 'p' is tunable and does not default to using the L2-norm. Instead, it utilizes a validation set to select the appropriate type of norm. However, the benefits of this process are challenging to quantify, and it's unclear how impactful they may be. The authors have not conducted a corresponding ablation study to demonstrate that the chosen norm is indeed superior to a p optimized through selection. 2) The referenced paper introduces logit normalization during training-time, whereas this study applies it as a post-hoc measure. The amount of novelty this aspect brings is somewhat limited.

Quality: There's inconsistency in the performance metrics used in the experiments, with initial methods based on NAURC and subsequent ones relying on SAC, without offering a comparative analysis across all metrics in the appendix. This lack of uniformity casts doubt on the results' consistency and the experiment's overall reliability.

Clarity: The paper could be significantly improved by providing more intuition/motivation and explaining the rationale behind methodological choices. For instance, the reasons why p-norm would perform effectively in selective classification are not well-articulated. The paper references Wei et al. (2022) but does not provide adequate context for readers less familiar with this work, compromising the clarity of the research.

Significance: The research's significance is limited, as it doesn't introduce considerable challenges beyond what is already known from the application of logit normalization to selective classification. Furthermore, intriguing findings, like those depicted in Figure 11, are not sufficiently highlighted or explained in the main text, representing a missed opportunity to underscore potential novel insights. The reasons behind the inability of high-confidence models to enhance selective classification remain unexplored, leaving gaps in understanding.

**Questions:**

1) What is the intuition behind using p-normalization? Why is logit information helpful in distinguishing between correctly classified and incorrectly classified samples?
2) For figure 10:  “models that produce highly confident MSPs tend to have better confidence estimators (in terms of NAURC), while models whose MSP distribution is more balanced tend to be easily improvable by post-hoc optimization—which, in turn, makes the resulting confidence estimator concentrated on highly confident values.” In instances where logit normalization proves ineffective, could the issue stem from the original model being under-calibrated, blurring the distinction between correctly and incorrectly classified samples due to a minimal confidence gap? My guess of why the logit normalization works is that: it is actually making the correct sample have much higher confidence while incorrect samples have slightly lower confidence, and therefore the confidence can distinguish them well. If this is the case, would it be possible to consider the corresponding ECE (compared to the histogram) to see if under-confident models are more easily improved, while over-confident models are not?  Based on my observation, many models in timm are actually under-confident due to insufficient training.
3) Why is learnable p-norm better than L2-norm? Why is a different norm required for different datasets? How significant is the benefit of this learnability? I think an ablation study can be better to support the use of learnable p.
4) What loss objective was used in the training of 'p' and 't'? Was it AURC or normalized AURC?
5) How was MSP FALLBACK considered in the experiments? Was it only taken into account when calculating the evaluation metric "average positive gain"? For a dataset, what criteria should we use to decide whether to use MSP or logit normalization? Should we still rely on its performance on the validation set, or is there a separate held-out set for deciding whether to use MSP Fallback?
6)  Is the third graph in Figure 3 also at 0.3491? It seems to have a high correlation.
7) Can you provide a more detailed explanation of Figure 3? What kind of message can we read from different optimal values of 'p'? Are models with different 'p' but the same architecture shown in the same plot?
8) In Sec 4.3, it states: "This robustness can be explained by a strong correlation between the selective performance in the original test set and under distribution shift." Could you elaborate on what this sentence means?
9) If time permits, it would be better to include a result for NAURC in Table 4 for consistent comparison.

---

> ### Author Response · Authors · 2023-11-19
> **Response to Reviewer aDAg (1)**
>
> Thank you very much for your constructive feedback. Please find our responses below. We hope they are satisfactory and you will consider raising your score.
>
> >**W1/Q3:** Why is learnable p-norm better than L2-norm? Why is a different norm required for different datasets? How significant is the benefit of this learnability? I think an ablation study can be better to support the use of learnable p.
>
> Thank you for this suggestion. We have added a new Appendix G with an ablation study regarding the choice of p. As shown in the table below, for MaxLogit-pNorm, a tunable p achieves much higher gains than a fixed p=2. For MSP-pNorm, the benefits (of using a tunable p rather than a fixed p=2) are smaller, although still statistically significant (see the Appendices F and I of the new version). However, it should be noted that, differently from MaxLogit, using MSP with p-norm normalization always requires the tuning of the temperature (otherwise the performance deteriorates). This additional tuning, besides being less computationally efficient (since more values need to be evaluated), causes worse data efficiency, as shown in Appendix F (note that we added the data efficiency curve for MSP-pNorm with fixed p=2).
>
> |$p$|MaxLogit-pNorm|MSP-pNorm|
> |-|-|-|
> |0|0.00000|0.05776|
> |1|0.00197|0.06030|
> |2|0.01471|0.06501|
> |3|0.05039|0.06714|
> |4|0.06416|0.06798|
> |5|0.06778|0.06787|
> |6|0.06785|0.06741|
> |7|0.06654|0.06703|
> |Tunable|0.06837|0.06781|
>
> >**W2:** The referenced paper introduces logit normalization during training-time, whereas this study applies it as a post-hoc measure. The amount of novelty this aspect brings is somewhat limited.
>
> Concerning specifically logit normalization, we would like to emphasize the following points about the novelty of this aspect of our work:
>
> * (Wei et al., 2022) proposed normalization in the context of calibration and OOD detection. We propose logit normalization to improve selective classification performance. These are different tasks whose performance is not aligned, e.g., improving calibration may harm selective classification and vice-versa.
> * (Wei et al., 2022) only used logit normalization during training time; during inference, they still used the raw MSP as confidence estimator, without any normalization. It is currently unknown if this would bring any benefits to selective classification. In contrast, we propose to use it during inference, which can be easily applied on top of any trained model without requiring modifications to its training regime.
> * (Wei et al., 2022) used only the L2 norm, while we propose using a general p-norm. This extension is not only significant, as shown in the new Appendices F, G, and I, but also theoretically grounded, as shown in the new Appendix C. To the best of our knowledge, applying logit p-norm normalization had not been proposed before in any context.
> * (Wei et al., 2022) used what we would call “MSP-pNorm with fixed p=2”, which still requires tuning a temperature hyperparameter due to the use of a softmax function. In contrast, we have shown that applying logit p-norm normalization followed by MaxLogit not only achieves a better performance, but also avoids the temperature tuning, resulting in better stability and data efficiency (see the new Appendices F and I).
> * There is a gap in the literature with respect to the choice of the best post-hoc confidence estimator for selective classification, with many previous works using the MSP as baseline. Even if MSP-pNorm with fixed p=2 turned out to be the best method in our benchmark, performing such a benchmark is (we believe) a significant contribution to this literature.
>
> Please note that, besides these, there are also several other original aspects in our work:
>
> * We solve an open problem reported in (Galil et al., 2023), which is to fix the anomaly in selective classification observed in many classifiers. In particular, we show that this is not an intrinsic limitation of these classifiers, but simply a consequence of using the MSP; when a better confidence estimator is used (which can be easily found with a few evaluations on hold-out validation data), the anomaly disappears.
> * We investigate the robustness to distribution shift, both of the proposed post-hoc fix as well as of selective classification itself (after the post-hoc fix). As far as we know, our paper is the first to explore this aspect.

---

> ### Author Response · Authors · 2023-11-19
> **Response to Reviewer aDAg (2)**
>
> >**W3/Q9:** There's inconsistency in the performance metrics used in the experiments, with initial methods based on NAURC and subsequent ones relying on SAC, without offering a comparative analysis across all metrics in the appendix. If time permits, it would be better to include a result for NAURC in Table 4 for consistent comparison.
>
> Thank you for this observation and suggestion. We have added the NAURC results for all models in Section 4.3 (see the new Figure 4). Moreover, we also added to a new Appendix D.5 the SAC results for all models, which reinforces our conclusions of Section 4.2. The reason to use different metrics is the following: when comparing confidence estimators across different models and/or test datasets (which is the main goal of this paper), the NAURC is an appropriate metric as argued in Section 3.2. However, after the best confidence estimator is chosen for each model and we wish to compare models, then other metrics are potentially more meaningful; that is why we include AURC (and now SAC) results in Section 4.2. Moreover, the SAC is arguably a more tangible metric, that is why we wanted to include it towards the end of our paper, to give a more concrete illustration of the impact of selective classification (note that we significantly rewrote Section 4.3 to make this all clear). Nevertheless, we should emphasize that, in all our experiments, the results are consistent across all metrics.
>
> >**W4:** The paper could be significantly improved by providing more intuition/motivation and explaining the rationale behind methodological choices. For instance, the reasons why p-norm would perform effectively in selective classification are not well-articulated. The paper references Wei et al. (2022) but does not provide adequate context for readers less familiar with this work, compromising the clarity of the research.
>
> We have included a new Appendix C (and a pointer to it on Section 3.1.2) containing appropriate context on logit normalization during training as proposed by Wei et al. (2022), as well as a detailed explanation of our intuition and theoretical justification for p-norm normalization as a post-training technique. In summary, while our original intuition was that using logit normalization post-training would have a similar behavior to using it during training, our experimental results (Appendix J of the new version) actually disproved this hypothesis. We now have been able to articulate a partial explanation, showing that logit p-norm normalization (with a tunable p) is closely related to temperature scaling (where p acts as 1/T). We hope the reviewer is satisfied with our new material.
>
> >**W5:** The research's significance is limited, as it doesn't introduce considerable challenges beyond what is already known from the application of logit normalization to selective classification. Furthermore, intriguing findings, like those depicted in Figure 11, are not sufficiently highlighted or explained in the main text, representing a missed opportunity to underscore potential novel insights. The reasons behind the inability of high-confidence models to enhance selective classification remain unexplored, leaving gaps in understanding.
>
> As explained in our response to W2, our paper contains considerable novelty beyond what is already known, and not only in aspects related to logit normalization, but also in selective classification in general and its robustness to distribution shift. Regarding the intriguing findings mentioned by the reviewer, we were not able to elaborate further in the main text due to lack of space, but we have now included additional insights in Appendix C. Nevertheless, the fact that our paper poses a new open problem should not, in our opinion, constitute grounds for rejection; on the contrary, we believe this difficult question is intriguing enough to merit reporting, and we hope to inspire other researchers to investigate it in the future.
>
> >**Q1:** What is the intuition behind using p-normalization? Why is logit information helpful in distinguishing between correctly classified and incorrectly classified samples?
>
> Using logit information is natural since it is readily available for any softmax-based classifier and the widely used MSP is itself a function of the logits. For this reason many previous works have considered functions of the logits to try to improve upon this baseline, across many related problems such as misclassification detection, calibration and OOD detection. Using logit information, possibly with a few tunable parameters, is also the simplest possible approach, so in this paper our goal was to extensively benchmark these techniques in order to establish a strong baseline for future research. Regarding the former question, we have included a new Appendix C with the intuition behind p-norm normalization, as explained in our response to W4.

---

> ### Author Response · Authors · 2023-11-19
> **Response to Reviewer aDAg (3)**
>
> >**Q2:** In instances where logit normalization proves ineffective, could the issue stem from the original model being under-calibrated, blurring the distinction between correctly and incorrectly classified samples due to a minimal confidence gap? My guess of why the logit normalization works is that: it is actually making the correct sample have much higher confidence while incorrect samples have slightly lower confidence, and therefore the confidence can distinguish them well. If this is the case, would it be possible to consider the corresponding ECE (compared to the histogram) to see if under-confident models are more easily improved, while over-confident models are not?
>
> We believe the reviewer means “In instances where logit normalization proves **effective**” and that “under-calibrated” means the same as “underconfident”. While we would not have time now to plot calibration curves (not simply ECE as the issue here is underconfidence), the results mentioned by the reviewer (Appendix J of the new version) already confirm this hypothesis: since all of these ImageNet models have relatively high accuracy, whenever their MSP histogram is concentrated on relatively low values, it follows that the model is underconfident. This happens essentially for all models where logit normalization (as well as temperature scaling) is effective, i.e., for the models which have relatively poor performance on selective classification using the MSP. Whether these models are underconfident due to insufficient training or some other reason, it is a question that we hope to address in our future work as well as inspire other researchers to investigate.
>
> >**Q4:** What loss objective was used in the training of 'p' and 't'? Was it AURC or normalized AURC?
>
> We used AURC as mentioned in Section 3.1.2. However, note that, when the classifier is fixed and we are evaluating some parameter of a confidence estimator, NAURC is a strictly monotonic function of AURC, since it is a simple min-max scaling (see Section 3.2.2), so optimizing AURC is equivalent to optimizing NAURC.
>
> >**Q5:** How was MSP FALLBACK considered in the experiments? (...) For a dataset, what criteria should we use to decide whether to use MSP or logit normalization?
>
> The MSP fallback was only used to compute APG and whenever explicitly mentioned in the caption of a figure. Assuming the original dataset is split into training (used for training the classifier and never used in this paper), validation and test sets, we should use the validation set to tune the hyperparameters of a confidence estimator. The MSP fallback is then interpreted as a binary hyperparameter, or more generally as an additional configuration to our hyperparameter search grid, e.g., {p=4, p=5, p=6, p=7, MSP}, as illustrated in Figure 3.
>
> >**Q6:** Is the third graph in Figure 3 also at 0.3491? It seems to have a high correlation.
>
> Indeed there was a plot error, thank you for pointing this out. The correct value for the correlation in this graph is -0.9019.
>
> >**Q7:** Can you provide a more detailed explanation of Figure 3? What kind of message can we read from different optimal values of 'p'? Are models with different 'p' but the same architecture shown in the same plot?
>
> Each plot in Figure 3 shows the 84 different classifiers considered, with no repetition in the same plot. For each model, the optimal value p is the value of p that optimizes the MaxLogit-pNorm confidence estimator (with MSP fallback, as indicated in the legend). Then, each subfigure shows these models before and after post-hoc optimization, i.e., the first and third plots (baseline) actually show the models using the MSP baseline, while the second and fourth plots show these same models after optimization. We have revised the caption of the figure to clarify this point. Roughly speaking, the figure shows that, before optimization, the selective classification performance of many classifiers is bad and all over the place, while after optimization it improves significantly and becomes much better correlated with accuracy, which makes sense. Thus, “broken” confidence estimators have been fixed.
>
> >**Q8:** In Sec 4.3, it states: "This robustness can be explained by a strong correlation between the selective performance in the original test set and under distribution shift." Could you elaborate on what this sentence means?
>
> The sentence was indeed confusing; thanks for pointing it out. We have significantly rewritten Section 4.3 to make all our points clearer. The corresponding part now reads: “First, we evaluate the performance of MaxLogit-pNorm on ImageNet and ImageNetV2 for all classifiers considered. Figure 4a shows that the NAURC gains (over the MSP baseline) obtained for ImageNet translate to similar gains for ImageNetV2, showing that this post-hoc method is quite robust to distribution shift.”
>
> We hope we have satisfactorily addressed all of the reviewer’s concerns and would be happy to answer any other questions.

---

> ### Author Response · Authors · 2023-11-21
> **Gentle reminder**
>
> Thank you again for your constructive feedback. We believe we have addressed all your concerns.
>
> We would like to remind you that discussion phase is ending soon. We hope you will consider raising your score. If there are still any reasons you think this work should not be accepted, please let us know and we will be happy to address them.

---

> > ### Comment · Reviewer_aDAg · 2023-11-22
> > **Thanks for the detailed response**
> >
> > Thank you for the very detailed response. I appreciate that this paper provides a simple alternative to temperature scaling and demonstrates its effectiveness. My hesitation stems from the fact that logit normalization has already been presented as an independent technique in previous research. While the authors argue that it hasn't been studied in the context of test-time selective classification tasks, in general, selective classification, OOD detection, and misclassification detection are closely uncertainty-estimation-related tasks. So the application of logit normalization to this context doesn't come as a surprise to me. Nevertheless, I am willing to adjust my score to a 6.

---

### Official Review · Reviewer_L98u · 2023-10-31

**Soundness:** 3 good
**Presentation:** 3 good
**Contribution:** 3 good
**Rating:** 6
**Confidence:** 4

**Summary:**

The paper considers the selective classification setup using a set of established post-hoc SC techniques. To improve performance, an additional optimization process deliberately optimizes SC method hyper-parameters based on a validation set. A new metric is proposed which fixes previous methods' sensitivity to the underlying classifier's risk. The post-hoc optimization process improves confidence estimation performance across a wide range of models. The paper also discusses selective classification performance under distribution shift, showing that SC performance degrades under stronger shifts and that better in-distribution performance correlates with stronger out-of-distribution performance.

**Strengths:**

- The paper addresses a timely and important topic in uncertainty quantification and trustworthy machine learning.
- The background section is very well written and easy to follow.
- The newly proposed normalized AURC metric seems to be an appropriate fix for accuracy sensitivity of the previously introduced metrics. It is important to properly evaluate selective classifiers and compare them meaningfully against each other.
- The post-hoc fix for confidence estimators seems promising.
- A lot of additional experiments and background is presented in the appendix.

**Weaknesses:**

- As per my understanding, the paper proposes to optimize hyper-parameters of selective classification methods based on a hold-out dataset. This process is never formally defined in the paper. Moreover, the paper does not consider potential (negative) side effects of this optimization procedure. Especially under distribution shift, deliberate calibration towards an in-distribution validation dataset might be more harmful than simply performing selective classification with a pre-defined threshold.
- Neither temperature scaling nor logit normalization are new concepts but have been shown in past work to help reduce overconfidence and to improve calibration. Therefore, the paper does not provide novelty in terms of new selective classification approaches.
- It is unclear to me what the take-away message from Tables 1 and 2 are. Based on these results, it is unclear when we expect pNorm to work and whether architecture (and if so, which exact part of it) do prevent us from improving performance with pNorm.
- Section 4.2 suggests that the described post-hoc SC methods can fix "broken" confidence estimators. As per my understanding, NAURC was introduced to remove the E-AURC's accuracy sensitivity. If the proposed post-hoc fix works, would that not remove the the need for the NAURC score as accuracy now determines SC performance (i.e., accuracy and calibration are correlated)?
- Section 4.3 which talks about performance under distribution shift. It appears like the finding that better in-distribution classification performance leads to out-of-distribution performance is also not new and discussed in [1]. Although the paper does provide the exact numbers of how strongly performance degrades, the take-away message beyond the effect already introduced in [1] appears limited.

**References**:

[1] Miller, John P., et al. "Accuracy on the line: on the strong correlation between out-of-distribution and in-distribution generalization." International Conference on Machine Learning. PMLR, 2021.

**Questions:**

Embedded in **Weaknesses** above.

I am willing to increase my score as part of the discussion phase if the authors can address my concerns.

---

> ### Author Response · Authors · 2023-11-15
> **Response to Reviewer L98u (1)**
>
> Thank you very much for your constructive feedback. Please find our responses below.
>
> > As per my understanding, the paper proposes to optimize hyper-parameters of selective classification methods based on a hold-out dataset. This process is never formally defined in the paper.
>
> We kindly request the reviewer to take a look at the first paragraph of section 3.1 (Tunable Logit Transformations), where our hyperparameter optimization process is described: "The idea is to take any parameter-free logit-based confidence estimator, such as those described in section 2.2, and augment it with a logit transformation parameterized by one or a few hyperparameters, which are then tuned (e.g., via grid search) using a labeled hold-out dataset not used during training of the classifier (i.e., validation data). Moreover, this hyperparameter tuning is done using as objective function not a proxy loss but rather the exact same metric that one is interested in optimizing, for instance, AURC or AUROC."
>
> Specific details on the optimization of T and p used for our empirical results are given in Appendix D.1 (mentioned in the last paragraph of section 4.1). The size of the corresponding hold-out set used is also mentioned whenever a dataset is introduced (Section 4 and Appendix E).
>
> > Moreover, the paper does not consider potential (negative) side effects of this optimization procedure. Especially under distribution shift, deliberate calibration towards an in-distribution validation dataset might be more harmful than simply performing selective classification with a pre-defined threshold.
>
> We kindly request the reviewer to take a look at the first paragraph of section 4.3 (Performance under Distribution Shift), where this important question is addressed. Our goal is to investigate whether optimizing a post-hoc confidence estimator using the in-distribution data harms selective classification (SC) performance under distribution shift. In Table 3, we show using the SAC metric (maximum coverage allowed for a given target accuracy, in this case chosen as 80.86%, the original accuracy without distribution shift) that this optimization is not only **not** harmful under distribution shift, it is actually beneficial, since the coverage drops less than without post-hoc optimization.
>
> > Section 4.3 which talks about performance under distribution shift. It appears like the finding that better in-distribution classification performance leads to out-of-distribution performance is also not new and discussed in [1]. Although the paper does provide the exact numbers of how strongly performance degrades, the take-away message beyond the effect already introduced in [1] appears limited.
>
> Please note that the paper [1] provides empirical results for a specific performance metric, namely, full-coverage accuracy. It is generally an open question whether the same correlation holds for other performance metrics, such as calibration and SC metrics. Indeed, as mentioned by the reviewer in the previous point, it is **not true** that an intervention that improves in-distribution calibration will also tend to improve out-of-distribution calibration. But in section 4.3 we show that gains from post-hoc optimization in in-distribution SC performance **do** translate to gains in out-of-distribution SC performance, a result which does not follow from [1].
>
> Additionally, considering all models after post-hoc optimization, we investigate in the revised section 4.3 (also in the new Appendix K) whether SC performance itself (as measured by NAURC) is robust to distribution shift. More precisely, we noticed that the SC performance of a classifier degrades under distribution shift; however, the (full-coverage) accuracy also degrades. It turns out that the increase in NAURC is entirely explained by the accuracy drop. This is not obvious: it is conceivable that distribution shift could produce an even higher degradation to NAURC (which would manifest itself by an RC curve that is higher at low coverage, even if we “scale” the curve to normalize for accuracy).
>
> Given the above points, we kindly request the reviewer to reconsider their opinion about the contributions of this section.

---

> ### Author Response · Authors · 2023-11-15
> **Response to Reviewer L98u (2)**
>
> > Neither temperature scaling nor logit normalization are new concepts but have been shown in past work to help reduce overconfidence and to improve calibration. Therefore, the paper does not provide novelty in terms of new selective classification approaches.
>
> In principle, the goal of the paper was not to propose new methods for selective classification, but rather to benchmark the existing methods across a large number of classifiers, instead of just a few classifiers as done in most papers. We considered a total of 20 methods by combining a logit-based confidence estimator with a tunable logit transformation, which were then applied to 84 ImageNet classifiers. Such an extensive study has never been done before.
>
> However, in the course of performing this benchmark study, we could not avoid making some contributions to selective classification methods. Namely:
> * We propose to apply logit normalization as a post-hoc method, while the original method was proposed to be applied during training;
> * We propose logit normalization with a general p-norm, while the original logit normalization only considered p=2. To the best of our knowledge, this extension to logit normalization had not been proposed before;
> * We propose to tune the confidence estimator (either the temperature or the p-norm parameter or both) using directly the AURC, which is also a novel idea. Note that conventional TS is optimized using the NLL;
> * Combining every logit-based confidence estimator with every applicable tunable transformation (among the ones we considered) produced several methods which had not been proposed before. Indeed, the best method found in our evaluation is a combination of MaxLogit and pNorm, which is not obvious and was entirely unexpected to us. Note that this differs significantly from applying logit normalization together with the softmax function as originally proposed in (Wei et al., 2022), which additionally requires temperature optimization (nor required by MaxLogit).
>
> Moreover, after picking the best post-hoc confidence estimator method for each classifier, our extensive evaluation enabled us to benchmark models. One of the conclusions—which to best of our knowledge is entirely novel in the selective classification literature—is that, across models, after post-hoc optimization, there is no tradeoff between selective and full-coverage accuracy, i.e., the best selective accuracy is obtained simply by the model with the best accuracy.
>
> Given the above points, we hope the reviewer agrees that the paper provides significant novelty to the selective classification literature.
>
> > It is unclear to me what the take-away message from Tables 1 and 2 are. Based on these results, it is unclear when we expect pNorm to work and whether architecture (and if so, which exact part of it) do prevent us from improving performance with pNorm.
>
> These 2 tables together (now unified as Table 1) show that:
> - an improvement is possible for some but not all classifiers;
> - when an improvement is possible, some confidence estimation methods work better than others.
> However, to really find out the best method (among the ones considered), we have to compute their average performance across all classifiers, which is done in Table 2, but we have to be careful to perform a fair comparison. Thus, Table 1 also serve to give an empirical motivation to the methodology described in section 3.2 (Evaluation Metrics).
>
> From these results (together with the data efficiency results in Appendix F), MaxLogit-pNorm emerges as a winner. However, these are empirical results. It is an open question to identify exactly which aspect of a model architecture or training regime renders it improvable (or non-improvable) by post-hoc optimization in general (or pNorm in particular). We believe this is an interesting question for future work that might inspire other researchers. Nevertheless, from the point of view of choosing a confidence estimation method for a particular classifier, it is a simple matter of experimenting to see whether it improves performance or not, similarly to optimizing a hyperparameter (as discussed in section 3.2.2), but with the benefit that it can be easily applied post-hoc, without retraining.

---

> ### Author Response · Authors · 2023-11-15
> **Response to Reviewer L98u (3)**
>
> > Section 4.2 suggests that the described post-hoc SC methods can fix "broken" confidence estimators. As per my understanding, NAURC was introduced to remove the E-AURC's accuracy sensitivity. If the proposed post-hoc fix works, would that not remove the the need for the NAURC score as accuracy now determines SC performance (i.e., accuracy and calibration are correlated)?
>
> The NAURC metric is useful whenever we wish to make comparisons of SC performance across different problems (where a problem includes the trained classifier, its confidence estimator and the test dataset). So it is a useful metric to evaluate confidence estimators while averaging performance across several classifiers, which is what we do in Section 4.1. It is also a useful metric to help us visualize the gains of a confidence estimator over the MSP baseline, which is what we do in Fig. 3(a) of Section 4.2. Now, if we are instead evaluating *classifiers*, each of them with the proposed post-hoc fix, then the reviewer is correct that in this case the AURC versus accuracy curve in Fig. 3(b) is probably more useful. Nevertheless, the NAURC could become useful again even in this case if we wish to compare classifier architectures while averaging SC performance across different datasets.
>
> (Since the reviewer mentioned calibration, we would like to remind that selective classification is not equivalent to calibration: as shown in previous work, there are cases where improving calibration harms selective classification and vice-versa.)
>
> We hope these responses address the reviewer's concerns and will be happy to answer any other questions.

---

> ### Author Response · Authors · 2023-11-21
> **Gentle reminder**
>
> Thank you again for your constructive feedback. We believe we have addressed all your concerns.
>
> We would like to remind you that discussion phase is ending soon. We hope you will consider raising your score. If there are still any reasons you think this work should not be accepted, please let us know and we will be happy to address them.

---

> > ### Comment · Reviewer_L98u · 2023-11-23
> > **Thank you**
> >
> > I thank the authors for their very detailed rebuttal. As a result, my concerns have been mostly addressed. I have increased my score accordingly.

---

### Author Response · Authors · 2023-11-21
**General response**

We would like to thank all the reviewers for their detailed and constructive feedback, which helped significantly improve our paper.

Below is a summary of the main concerns and how we have addressed them:

* **Insufficient comparison to post-training methods.** Originally we had included only methods that involved a tunable logit transformation. In the revised version, we included a comparison to other forms of tunable methods with a similar complexity (all that we could find), showing that our proposed method is still superior in performance while requiring much less hyperparameter tuning.
* **Why logit $p$-norm normalization with a tunable $p$ and not just $p=2$?** In the revised version, we performed an ablation study showing that a higher $p$ is beneficial compared to $p=2$. Moreover, we provided theoretical justification for a tunable $p$, showing that it is closely related to a certain form of temperature scaling.
* **Performance metrics.** All reviewers saw value in our proposed NAURC metric for the purposes of the evaluation done in this paper, however, reviewers L98u and 3CxY argued that other metrics may be more tangible when reporting the final performance of a selective classifier, with which we agree. For this reason we had already included one table with some results for the SAC metric, but reviewer aDAg saw this as creating inconsistency. To solve both concerns, we included complete results for both NAURC and SAC metrics, which are shown to be consistent with each other and to reinforce our conclusions.
* **Novelty.** As mentioned in the responses to reviewers L98u and aDAg, our paper contains a significant number of original contributions:
  1. We perform an extensive benchmark of post-hoc confidence estimation methods for selective classification, establishing strong baselines for future research. Such a study had never been done before and was arguably needed.
  2. In contrast to the original logit normalization, which uses the L2 norm and is applied only during training for a different problem, we propose it as a post-hoc method for selective classification, extend it to general $p$-norm, and consider a tunable $p$ with AURC as the optimization objective, all of which are new ideas.
  3. Moreover, instead of the usual MSP, we propose to combine logit $p$-normalization with MaxLogit, which dispenses with temperature scaling and leads to an exceptionally data efficient confidence estimator.
  4. We show that the proposed MaxLogit-pNorm is superior to existing methods, providing substantial (and statistically significant) gains in selective classification performance, which are preserved even under distribution shift.
  5. We show that our proposed method fixes anomalies reported in previous work, so that, after post-hoc optimization, selective classification performance becomes essentially determined by accuracy.
  6. We show evidence that selective classification performance itself is robust to distribution shift, in the sense that, although it naturally degrades, this degradation is not larger than what would be expected by the corresponding accuracy drop. As far as we know, our paper is the first to investigate selective classification under distribution shift.

The revised version contains the following new material:
* Additional NAURC results on Section 4.3 (Fig. 4) and Appendix K (Fig. 16);
* Additional SAC results on Appendix D.5 (Fig. 7);
* A new Appendix C with intuition and theoretical justification for logit $p$-norm normalization;
* Updated data efficiency figures (Appendix F);
* A new Appendix G with an ablation study on the choice of $p$;
* A new Appendix H with a comparison with other tunable methods;
* A new Appendix I on the statistical significance of the gains.

We believe we have addressed all the reviewers’ concerns and hope the reviewers will consider raising their scores.

---

### Meta-Review · Area_Chair_rKx1 · 2023-12-05

**Metareview:**

This paper carries out an empirical assessment of selective classification approaches focusing on ImageNet classifiers. For a number of pre-trained models, error detection approaches that operate on top of logits are benchmarked so as to elucidate differences in performance. Additional experiments are also carried out on top of other image classification datasets and some consistency is observed in terms of top performers, and situations are identified where a completely uninformative confidence score is made into a highly discriminative one upon tuning on held-out data. An evaluation metric is also introduced with the goal of reducing the influence of the prediction performance of the underlying classifier on its selective classification performance metrics.

While the research topic is highly relevant, the paper consists of an interesting empirical benchmark with conclusions all derived from empirical assessment, but we deemed the empirical methodology and scope to be lacking so that it's unclear to what extension conclusions hold in other settings and situations. To offer further detail, we list limitations within the empirical assessment that we believe require addressing.

In terms of scope, the empirical assessment is not broad enough to be conclusive and, while experiments on top of datasets other than ImageNet are reported in appendix E, the evaluation should cover other datasets and modalities, especially so considering that evaluations in this setting are relatively low cost given that training is only carried out for a few parameters on small samples. For instance, text classification tasks could have been considered. In other words, conclusions might change drastically if other types of data and model classes are considered. Another worth highlighting limitation in the scope is the lack of an analysis of calibration of the rejection scores under consideration. Metrics such as ECE should have been reported. Moreover, as pointed out by some of the reviewers, the set of chosen approaches is also not broad enough since the focus lies on simple transformations of logits.

In terms of methodology, except for appendix F, results are reported as aggregates as opposed to confidence intervals such that there are no significance levels to be accounted for and, since numbers are so close in many situations (cf. Table 1 for instance), conclusions could completely change under independent experiment runs due to, for instance, varying sets of held-out samples. We remark once again that the setting under consideration doesn't require compute intensive training and, as such, it's fair to expect rigorous empiricism and comparisons to be carried out in terms of significance levels rather than comparing aggregates.

Reviewers were all on the borderline. However, due to the limitations listed above, we conclude that the paper is not ready for publication at its current form. We encourage authors to expand their evaluation and review the analysis methodology for a more conclusive set of results.

**Justification For Why Not Higher Score:**

While the paper is interesting and tackles a very relevant problem, the evaluation requires improvements to provide clear evidence of the authors's claims.

**Justification For Why Not Lower Score:**

N/A

---

### Decision · Program_Chairs · 2024-01-16

Reject